Resource

# Atlas of amnion development during the first trimester of human pregnancy

Wenqi Hu ®[1], Carmen Sancho-Serra[2], Carlos W. Gantner ®[3],
Hanna M. Szafranska[1,3], Nita Solanky[4], Kate Metcalfe[4], Roser Vento-Tormo[2] &
Magdalena Zernicka-Goetz ®[1,3] ✉

The amnion is a critical extra-embryonic structure that supports foetal development, yet its ontogeny remains poorly defined. Here, using single-cell transcriptomics, we identified major cell types and subtypes in the human amnion across the first trimester of pregnancy, broadly categorized into epithelial, mesenchymal and macrophage lineages. We uncovered epithelial–mesenchymal and epithelial–immune transitions, highlighting dynamic remodelling during early pregnancy. Our results further revealed key intercellular communication pathways, including BMP4 signalling from mesenchymal to epithelial cells and TGF-β signalling from macrophages to mesenchymal cells, suggesting coordinated interactions that drive amnion morphogenesis. In addition, integrative comparisons across humans, non-human primates and in vitro stem cell-based models reveal that stem cell-based models recapitulate various stages of amnion development, emphasizing the need for careful selection of model systems to accurately recapitulate in vivo amnion formation. Collectively, our findings provide a detailed view of amnion cellular composition and interactions, advancing our understanding of its developmental role and regenerative potential.

The amnion is an extra-embryonic structure essential for the development of reptilian, avian and mammalian embryos as it encases the embryo, providing both mechanical and biochemical support. In humans, the amnion originates from a subset of pluripotent epiblast cells specified soon after implantation. Before implantation, the blastocyst consists of an outer extra-embryonic trophoblast layer and the inner cell mass that will differentiate into epiblast and extra-embryonic hypoblast by days 6–7 post-fertilization[1,2]. Upon implantation, epiblast cells polarize to form a rosette structure that undergoes lumenogenesis, creating the amniotic cavity as cells exit naive pluripotency[3–5]. Epiblast cells in contact with the hypoblast form the epiblast disc, which will develop into the embryo proper, while those in contact with the trophectoderm will form the amnion (Fig. 1a). The amnion not only

physically protects the embryo but also secretes essential hormones and cytokines that support embryonic development[6,7].

The human amnion is composed of two primary cell types—epithelial cells and mesenchymal cells—separated by a thick basement membrane[8,9]. Amniotic epithelial cells, which line the amniotic cavity, are responsible for producing the amniotic fluid, whereas the amniotic mesenchymal cells, embedded within the extracellular matrix, contribute to the structural scaffold of the avascular foetal membranes[10]. These membranes define the intrauterine cavity and protect the foetus during gestation[11].

The amnion undergoes extensive growth, repair and remodelling throughout pregnancy to align with embryonic development. These processes are closely associated with epithelial-to-mesenchymal

[1]Plasticity and Self-Organization Group, Division of Biology and Biological Engineering, California Institute of Technology, Pasadena, CA, USA. [2]Wellcome Sanger Institute, Wellcome Genome Campus, Cambridge, UK. [3]Mammalian Embryo and Stem Cell Group, Department of Physiology, Development and Neuroscience, University of Cambridge, Cambridge, UK. [4]Human Developmental Biology Resource, UCL GOS Institute of Child Health, London, UK. ✉e-mail: magdaz@caltech.edu

transitions (EMT) and mesenchymal-to-epithelial transitions (MET)[11–13]. In addition, EMT in the amnion has been reported to influence the immune properties of amniotic epithelial cells, often associated with localized inflammation and facilitating tissue remodelling[14].

Beyond its fundamental role in pregnancy, the human amnion serves as a valuable source of stem cells with multilineage differentiation potential. The stem cells derived from amnion can be utilized for cell-based therapies and regenerative medicine applications[8,15–18]. The unique properties of the amnion, including low immunogenicity[19], anti-inflammatory[20] and antimicrobial properties, make it an attractive candidate for various therapeutic applications[21], such as wound healing, treatment of ocular surface disorders and tissue engineering[22].

Despite growing interest, a comprehensive understanding of human amnion development remains limited. Single-cell RNA sequencing (scRNA-seq) of human pregastrulation embryos[23] and of primate gastrulating embryos[24,25] has provided transcriptional snapshots at specific stages of early development, offering preliminary insights into amnion development. Advances in stem cell technology, including the generation of stem cell-derived embryo-like models[25–27] and stem cell-derived amnion-like cells[28,29], have further clarified the developmental pathway of amnion specification. RNA sequencing (RNA-seq) of amnion tissue collected from pregnant women at term has confirmed the presence of multiple cell types within the fully developed amnion, including fibroblasts, epithelial cells, immunocytes and various intermediate cell types[30]. However, characterizing amnion from the early stages of human pregnancy remains challenging. In this study, we used scRNA-seq to profile various cell types present in human amnion during the first trimester of human pregnancy to gain insight into their interactions and potential functional contributions.

## Results

### Cell composition of human amnion in the first trimester

To explore the dynamics of transcriptional changes during amnion development, we collected seven human amnion samples representing 5–9 weeks of pregnancy and corresponding to Carnegie stages (CS) 16, 17, 19, 22 and 23, respectively. CS16 embryos have developed limb buds, the otic vesicle, early eye structures and the primitive heart tube, along with forming somites and the neural tube. CS17 embryos have developed hand rays, cartilage, ribs, intercostal muscles, mammary glands and the thymus. By CS19, embryos have developed the cerebral aqueduct, middle cerebral artery, renal artery and tibia. By CS22, the embryonic brain has developed nerve cell clusters and bundles of nerve fibres, and ossification has begun in the clavicle and long bones (Fig. 1a,b).

We prepared single-cell suspensions from four of these samples (CS16, CS17, CS19 and CS22) and performed scRNA-seq using the 10x Genomics Chromium system (Extended Data Fig. 1a). Cells with fewer than 500 or more than 8,000 genes expressed were excluded. In addition, we excluded cells with more than 20% mitochondrial reads to remove dead cells. Cell doublets were removed by Souporcell[31] analysis. In addition, amnion tissue can be contaminated during dissection with maternal cells such as maternal blood cells and blood vessels, we used Souporcell to analyse and remove cells that could be of maternal origin (Extended Data Fig. 1b). We also scored the cells and excluded those with marker gene expression characteristic of yolk sac, chorion, blood vessels and erythroid

cells (Extended Data Fig. 1c,d). In total, 14,027 single cells passed quality control and were included in our analysis. Data from the four stages were integrated using the 'IntegrateData' function in Seurat4[32].

Unsupervised clustering utilizing the Seurat package revealed ten distinct cell clusters defined by their transcriptional signatures (Extended Data Fig. 2a). We identified a total of six major cell types among the ten clusters based on their marker genes (Fig. 1c,d and Supplementary Table 1). These include amnion epithelial cells (AECs, clusters 1 and 5, marked by *GABRP* and *KRT18*), amnion mesenchymal cells (AMCs, cluster 0, marked by *MGP* and *VIM*), fibroblasts (cluster 3, marked by *COL6A1* and *COL5A1*), macrophages (cluster 9, marked by *MRC1* and *CD36*) and two clusters of actively proliferating cells (marked by *CDK1* and *TOP2A*), which were defined as amnion mesenchymal stem cells (AMSCs, clusters 2, 6 and 8) and amnion epithelial stem cells (AESCs, clusters 4 and 7) based on the expression of lineage-specific genes (Fig. 1d and Extended Data Fig. 2b). Their stem cell characteristics were also demonstrated by subsequent pseudotime analysis. The immunofluorescence staining of sectioned tissues confirmed the presence of epithelial cells (expressing E-Cadherin and KRT18), mesenchymal cells (expressing VIM), fibroblasts (expressing N-Cadherin), and macrophages (expressing CD45) in human CS19 and CS23 amnion tissues (Fig. 1e and Extended Data Fig. 3a–e).

### Cell subtypes and lineage trajectories in amnion development

To provide a more comprehensive and detailed depiction of the amnion's cellular composition, we used three-dimensional (3D) Uniform Manifold Approximation and Projection (UMAP) plots (Fig. 2a and Extended Data Fig. 4a). This approach allowed us to further subdivide the AESCs into two distinct groups, labelled as AESCs_1 (cluster 7) and AESCs_2 (cluster 4). Moreover, we identified a population of intermediate-state cells (cluster 5) that express both epithelial and mesenchymal marker genes (Fig. 2b and Extended Data Fig. 4b). In addition, our analyses revealed a group of cells with high expression of ectoderm markers such as *SOX2*, *TUBB3* and *NR2F1* (Fig. 2b and Extended Data Fig. 4c), which we labelled as amnion ectodermal cells (Amnion-Ect, AM-Ect). These ectoderm markers were also found to be expressed in the bulk RNA-seq of first and second trimester of human amnion samples[33] (Extended Data Fig. 4d) and in the scRNA-seq of CS8-11 cynomolgus monkey amnion cells[34] (Extended Data Fig. 4e,f). Immunofluorescence staining suggests the presence of amnion ectodermal cells (expressing SOX2 and TUBB3) in the CS16 and CS19 amnion section (Fig. 2c and Extended Data Fig. 4g).

To investigate the developmental trajectories within amnion cells, we utilized three pseudotime analysis methods. RNA velocity[35] revealed two primary trajectories: epithelial and mesenchymal. The epithelial trajectory further branched into two distinct paths: one transitioning from AESCs_1 to AESCs_2 and the other from AESCs_1 to AECs and then to macrophages. The mesenchymal trajectory delineated a progression from AMSCs to AMCs and finally to fibroblasts (Fig. 2d). Similarly to the RNA velocity results, trajectory and pseudotime analyses using Monocle3[36] and Destiny[37] also revealed the same two developmental trajectories (Fig. 2e,f).

Based on the UMAP and pseudotime analyses, AESCs_2 and intermediate cells may serve as bridges between epithelial and

**Fig. 1 | scRNA-seq analysis of human amnion during the first trimester.** **a**, A diagram illustrating the development of the human amnion. Trophectoderm is marked in grey, amnion in orange, extra-embryonic mesoderm in red, hypoblast or yolk sac endoderm in green, epiblast/embryo body in burgundy or pink and umbilical cord in purple. **b**, Images of human embryos representing different stages. CS16–CS19 were the collected sample, and CS22 is a representative image taken from the same centre. Arrows point to the amniotic membrane, and triangles mark the yolk sac. Scale bar, 0.5 cm. **c**, UMAP displaying the identified cell types within the analysed samples. **d**, A bubble plot showing selected top marker gene expression across cell types. **e,** Immunofluorescence (IF) staining

of amnion sections, with protein markers labelled at the top of the image and amnion stages labelled at the bottom. Different fluorescent markers highlight the localization of the proteins in the tissue. The region inside the white box is magnified on the right. White arrows indicate double-positive cells. Scale bars, 50 μm. CD45−VIM co-staining is representative of four independent experiments that yielded similar results. CD45−E-Cadherin co-staining is representative of two independent experiments. VIM−E-Cadherin and N-Cadherin−KRT18 are representative of one experiment. scRNA-seq analyses depicted in this figure are generated from human amnion samples of the following developmental stages: CS16 (*n* = 1), CS17 (*n* = 1), CS19 (*n* = 1) and CS22 (*n* = 1).

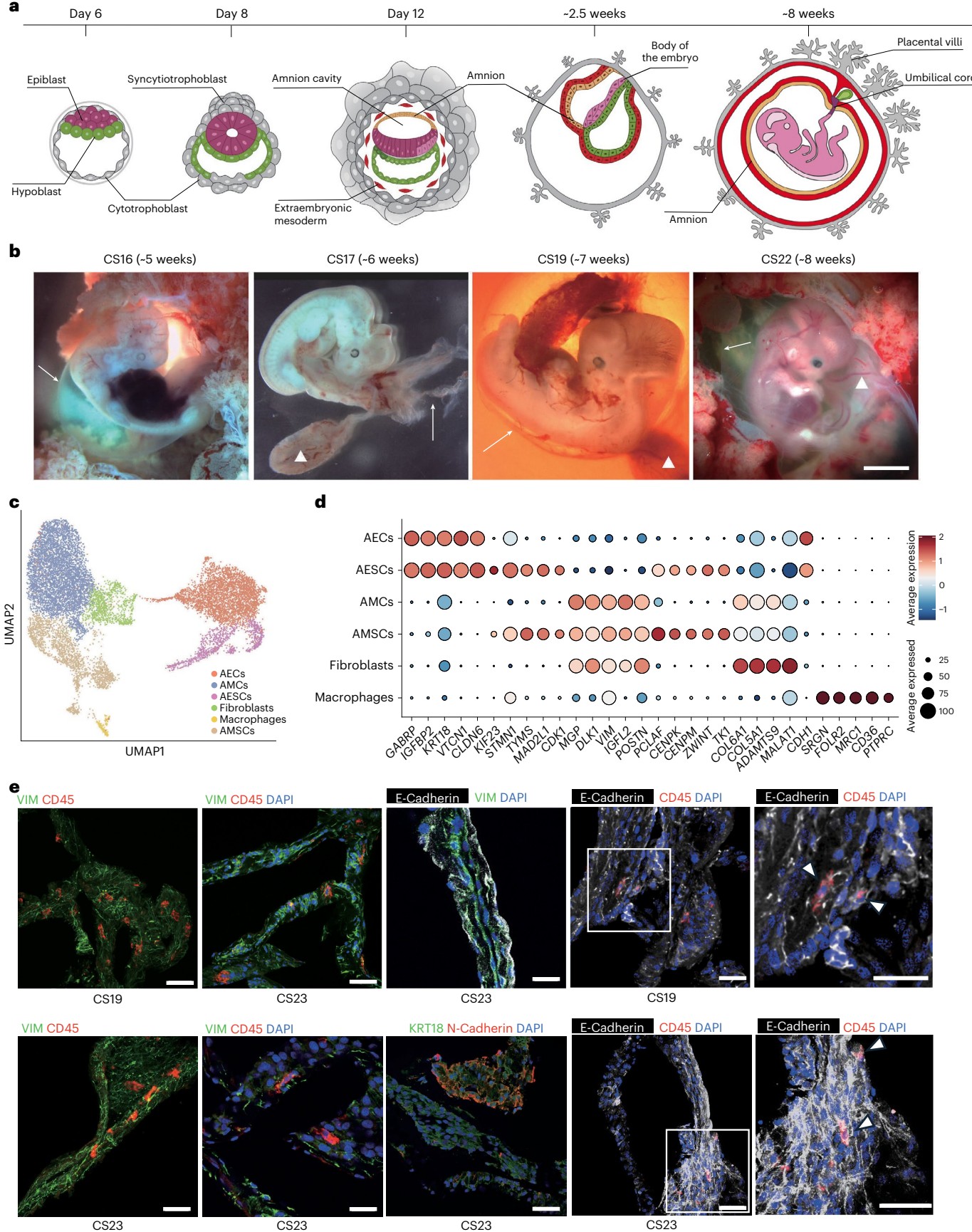

mesenchymal lineages, suggesting that an EMT occurs in the amnion. Consequently, we examined the expression of transcription factors related to EMT in different cell subtypes. Our analyses indicated that most EMT-related transcription factors were highly expressed in the mesenchymal lineage (Extended Data Fig. 5a). Specifically, expression of *SNAI1* and *SNAI2* was detected in the AESCs_2 cells, while expression of *SNAI2*, *ZEB1* and *TWIST1* was observed in the intermediate cells. Interestingly, we discovered high *SNAI1* expression only in the AESCs_2 cells that were closest to the mesenchymal lineage (Extended Data Fig. 5b), suggesting that AESCs_2 might be transitioning into mesenchymal cells through EMT.

Our analyses also identified transcripts that change expression in accordance with pseudotime (Supplementary Table 2). Specifically, along the epithelial–macrophage trajectory, we observed a progressive increase in the expression levels of epithelial and macrophage marker genes, such as *GABRP, IGFBP3, MRC1* and *CD36*. By contrast, mesenchymal marker genes such as *MGP* and *VIM* are systematically downregulated (Fig. 2g). Interestingly, this trend was reversed in the mesenchymal trajectory, where mesenchymal marker genes showed an increase in expression, highlighting the distinct and dynamic cellular behaviours in different developmental paths.

## Intercellular communication in amnion development

To investigate intercellular communication among amniotic cells, we utilized CellChat[38], which uncovered numerous potential interactions between various cell populations (Fig. 3a,b and Extended Data Fig. 6a). We examined the expression of receptors and ligands within cells to identify the roles of different cell types in the interaction network. We found that macrophages and intermediate cells exhibited an outgoing profile, primarily expressing ligands, whereas amnion ectodermal cells displayed an incoming profile, predominantly expressing receptors. Other cell types demonstrated a combination of both outgoing and incoming signalling capabilities (Fig. 3c).

We classified cells into three patterns based on their expression of genes encoding secreted signalling ligands (Fig. 3d,e and Supplementary Table 3). The epithelial pattern included AESCs_1, AESCs_2, AECs and intermediate cells, which expressed ligands associated with bone morphogenetic protein (BMP), Wingless/Integrated (WNT), platelet-derived growth factor (PDGF) and growth differentiation factor (GDF) signalling. The mesenchymal pattern consisted of AMSCs, AMCs and fibroblasts, which expressed ligands associated with midkine (MDK), non-canonical Wnt (ncWNT), hepatocyte growth factor (HGF) and insulin-like growth factor (IGF). Finally, the macrophage pattern showed expression of secreted phosphoprotein 1 (SPP1) and transforming growth factor beta (TFG-β).

Our analyses identified several growth factors that are linked to specific growth and developmental stages of amniotic cells, illustrating a complex intercellular communication network within the amnion. By analysing ligand and receptor expression, we were able to identify the likely signalling and responding cells across different cell populations (Fig. 4a–c). Specifically, our analyses indicated that within the BMP signalling pathway, which is known to be critical for amnion development[28,39,40], cells of the epithelial lineage function primarily as recipients of BMP signals (Fig. 4d). Interestingly, we found that *BMP4* was primarily expressed by amnion mesenchymal lineage, whereas

*BMP7* was predominantly expressed by the epithelial cells themselves (Fig. 4e and Extended Data Fig. 7a). The receptors for these proteins, *BMPR1A*, *ACVR2A*, *ACVR2B* and *BMPR2*, were mainly expressed in the epithelial lineage cells (Extended Data Fig. 7b). In addition, we also identified potential crosstalk between the MDK and WNT signalling pathways in amnion cells (Extended Data Fig. 7c–f). In our in vitro human stem cell differentiation experiments, BMP4 treatment of induced pluripotent stem (iPS) cells resulted in the upregulation of both early and late amnion markers (Extended Data Fig. 8a).

Some signalling pathways were also related to the characteristics of the amnion, including angiogenesis (PDGF, Fig. 4f), anti-inflammatory effects (IL6, Fig. 4g), EMT promotion (TGF-β, Fig. 4h) and immunosuppression (SPP1, Fig. 4i). The immunomodulatory properties of the amnion were particularly striking as we found that amnion cells expressed various immunosuppressive factors, such as macrophage migration inhibitory factor (*MIF*) (Extended Data Fig. 8b) and *SPP1* (Extended Data Fig. 8c), which play crucial roles in inhibiting immune responses[41–44]. Immunostaining of CS16 amnion sections confirmed the expression of MIF, SPP1 and its receptor CD44[41] in the amnion (Extended Data Fig. 8d). The expression of immunosuppressive factors provides a possible explanation for the amnion's capacity to inhibit immune responses. Overall, these pathways highlight the complexity of cellular communication and may play an integral role in coordinating cellular interactions during amnion development.

## Analysis of amnion development in vivo and in vitro

To investigate the developmental dynamics of amnion across different species and experimental conditions, we combined published scRNA-seq datasets from human CS7[24] (Extended Data Fig. 9a) with data from monkey CS8–11[34] (Extended Data Fig. 9b). In addition, we incorporated three sets of in vitro-derived amnion-like cells generated from stem cells[25,28,39] (Extended Data Fig. 9c–e) and included data from our own study on human amnion samples CS16–22.

By leveraging a combined UMAP analysis, we identified a clear developmental progression of amnion formation (Fig. 5a). To further delineate this trajectory, we isolated amnion cells from these datasets and performed a diffusion map analysis (Fig. 5b). Interestingly, our findings revealed that the different in vitro amnion models correspond different developmental stages in vivo. Specifically, the amnion models derived from human pluripotent stem (hPS) cells using either in a microfluidic device[28] or 3D biomimetic culturing[39] resembled earlier amnion stages (around CS7). By contrast, amnion-like cells derived from two-dimensional (2D) hPS cell cultures[25] more closely reflected later stages of amnion development (CS11–16) (Fig. 5c). This divergence highlights variations in developmental timing across different in vitro models, emphasizing the need for careful selection of model systems to accurately recapitulate in vivo amnion development.

Building on this integrative analysis, we performed differential expression and gene module analyses along the amnion developmental trajectory. Genes were categorized into distinct modules based on their expression patterns, and their average expression was visualized in a heatmap (Fig. 5d and Supplementary Table 4). We identified several transcription factors, including *POU5F1*, *HAND1* and *ISL1*, as being enriched in the early stages of amnion development. Gene Ontology (GO) annotations for these genes were predominantly associated

**Fig. 2 | Cell subtypes and lineage trajectories in human amnion. a**, A 3D UMAP representation showing different cell subtypes in the amnion. **b**, A violin plot showing marker gene expression across subtypes. **c**, Top: immunostaining of GABRP and SOX2 in the CS16 amnion section, with triangles marking the ectodermal cells. Representative image from two independent experiments. Bottom: immunostaining of E-Cadherin and TUBB3 in the CS19 amnion section. Scale bars, 50 μm. Representative image from four independent experiments. **d**, RNA velocity analysis indicating development tendencies of epithelial and mesenchymal cells, based on the integrated analysis of four independent

biological samples. **e**, Pseudotime and trajectory plots showing the epithelial–macrophage trajectory (top) and mesenchymal trajectory (bottom). **f**, Cell subtypes arranged along the Destiny pseudotime in two trajectories: epithelial–macrophage lineage (left) and mesenchymal lineage (right). **g**, The expression of selected differentially expressed genes (DEGs) during lineage progression in amnion: macrophage (top), epithelial (middle) and mesenchymal (bottom). scRNA-seq analyses depicted in this figure are generated from human amnion samples of the following developmental stages: CS16 (*n* = 1), CS17 (*n* = 1), CS19 (*n* = 1) and CS22 (*n* = 1).

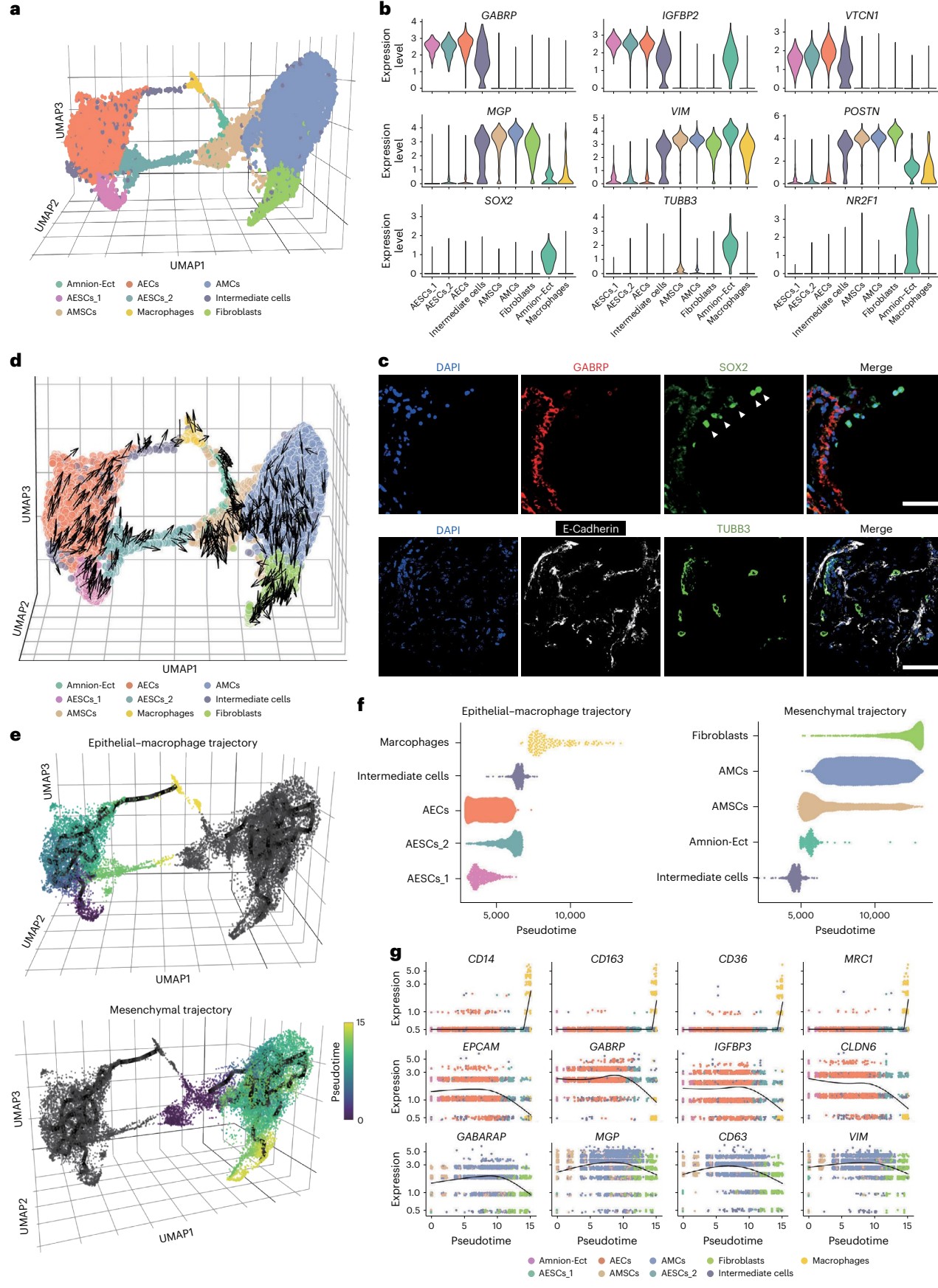

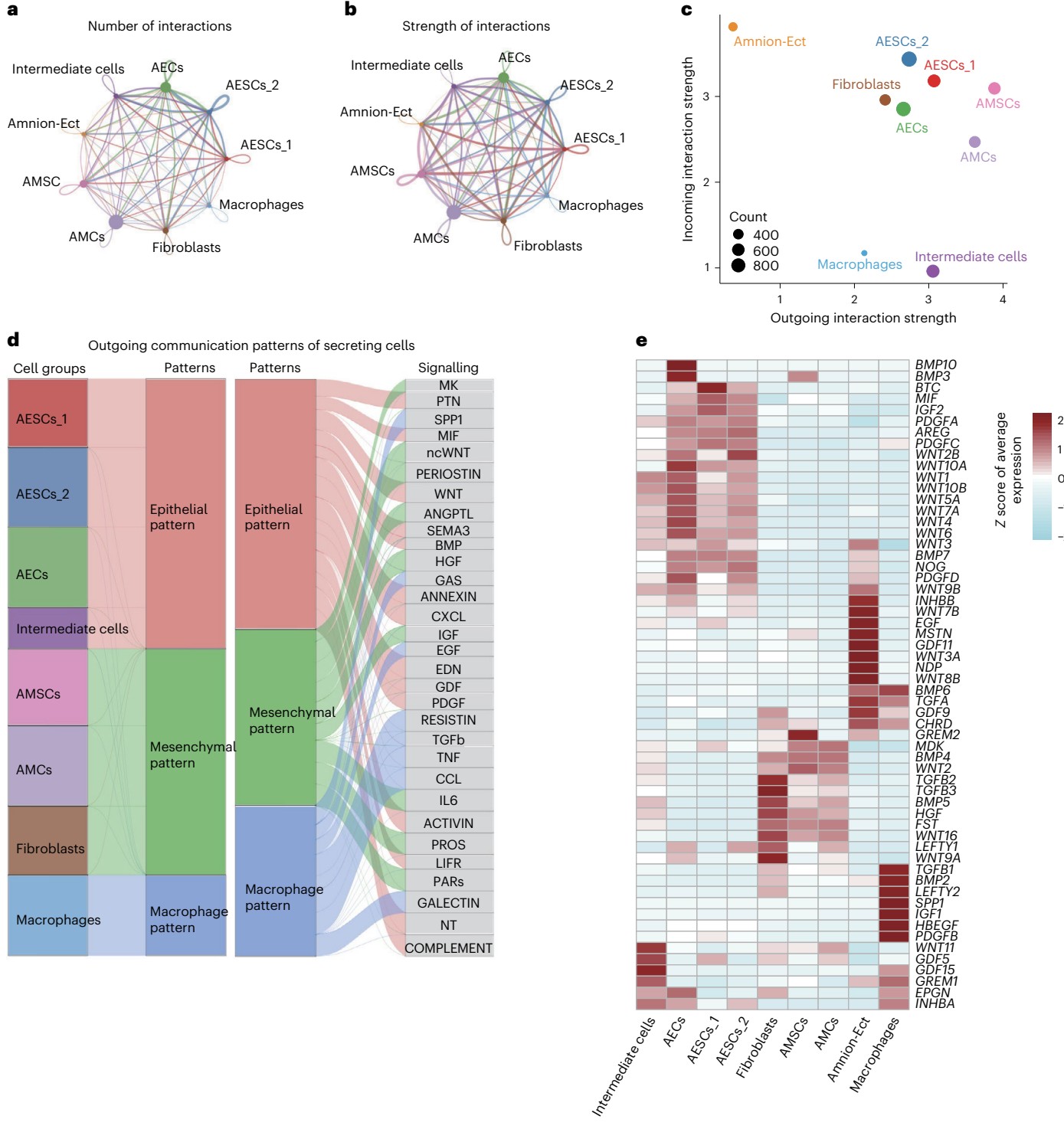

**Fig. 3 | Cellular communication and secretion patterns within the amnion.**
**a**, The total number of inferred signalling interactions among different cell types.
**b**, Overall interaction strength representing the cumulative communication probability between cell types. **c**, Cell roles in secreting and receiving signals. **d**, Classification of cells into three distinct secretion patterns based on gene expression profiles. **e**, A heatmap showing the expression of ligands across different cell types. scRNA-seq analyses depicted in this figure are generated from human amnion samples of the following developmental stages: CS16 ($n$ = 1), CS17 ($n$ = 1), CS19 ($n$ = 1) and CS22 ($n$ = 1).

with pathways such as 'embryonic organ development' and 'signalling pathways regulating pluripotency of stem cells' (Fig. 5e). By contrast, late-stage AMCs predominantly express mesenchymal markers such as *COL15A1* and *MGP*, along with transcription factors related to EMT, including *GATA6*, *TWIST1* and *ZEB1*. GO annotations for these genes were enriched in 'extracellular matrix' and 'mesenchyme development'

(Fig. 5f). In late-stage AECs, gene expression was enriched for epithelial markers such as *GABRP* and *IGFBP5*, as well as transcription factors *TFAP2A* and *TFAP2B*. GO annotations for these genes were associated with 'extracellular matrix' and 'cell adhesion molecule binding' pathways (Fig. 5g). This comprehensive genomic analysis illustrates the dynamic changes that occur throughout amnion development.

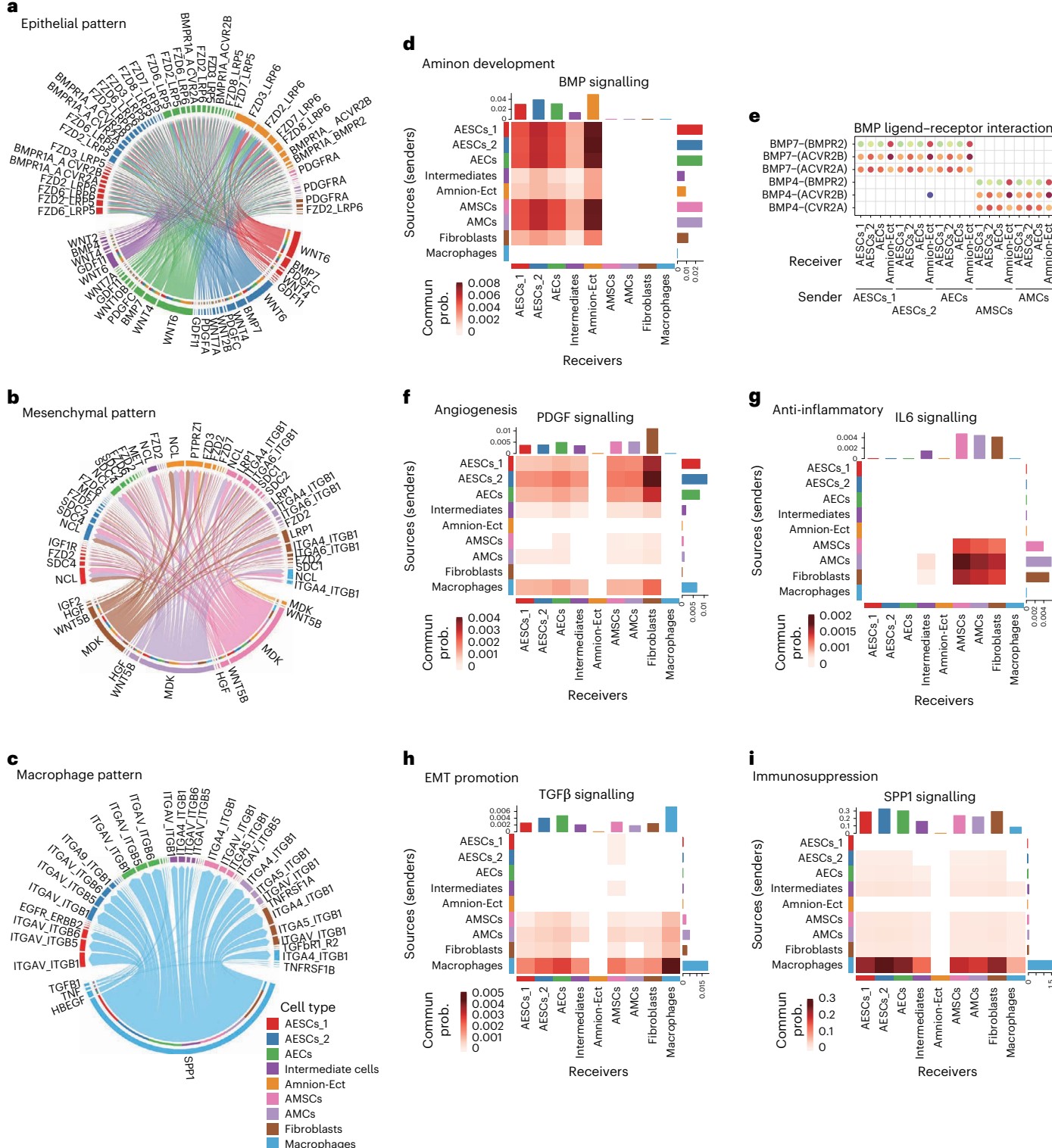

**Fig. 4 | Key signalling pathways in distinct amnion cell patterns. a–c,** A chord diagram showing ligand–receptor interactions in epithelial (**a**), mesenchymal (**b**) and macrophage (**c**) patterns. Distinct cell types are represented by different colours. **d,** A heatmap showing interactions in the BMP signalling pathway. Commun prob., communication probability. **e,** A bubble plot showing the significant interactions in the BMP signalling pathway. **f–i,** Heatmaps showing interactions in PDGF (**f**), IL6 (**g**), TGF-β (**h**) and SPP1 (**i**) signalling pathways. scRNA-seq analyses depicted in this figure are generated from human amnion samples of the following developmental stages: CS16 (*n* = 1), CS17 (*n* = 1), CS19 (*n* = 1) and CS22 (*n* = 1).

## Discussion

Our single-cell analysis of the human amnion has revealed a dynamic cellular landscape, developmental trajectories and intercellular interactions between different amnion cell types. Within the first trimester amnion, we identified six major cell types and nine cell subtypes spanning epithelial, mesenchymal and macrophage lineages. In addition, we also discovered a population of amnion ectodermal cells expressing some neural-related genes.

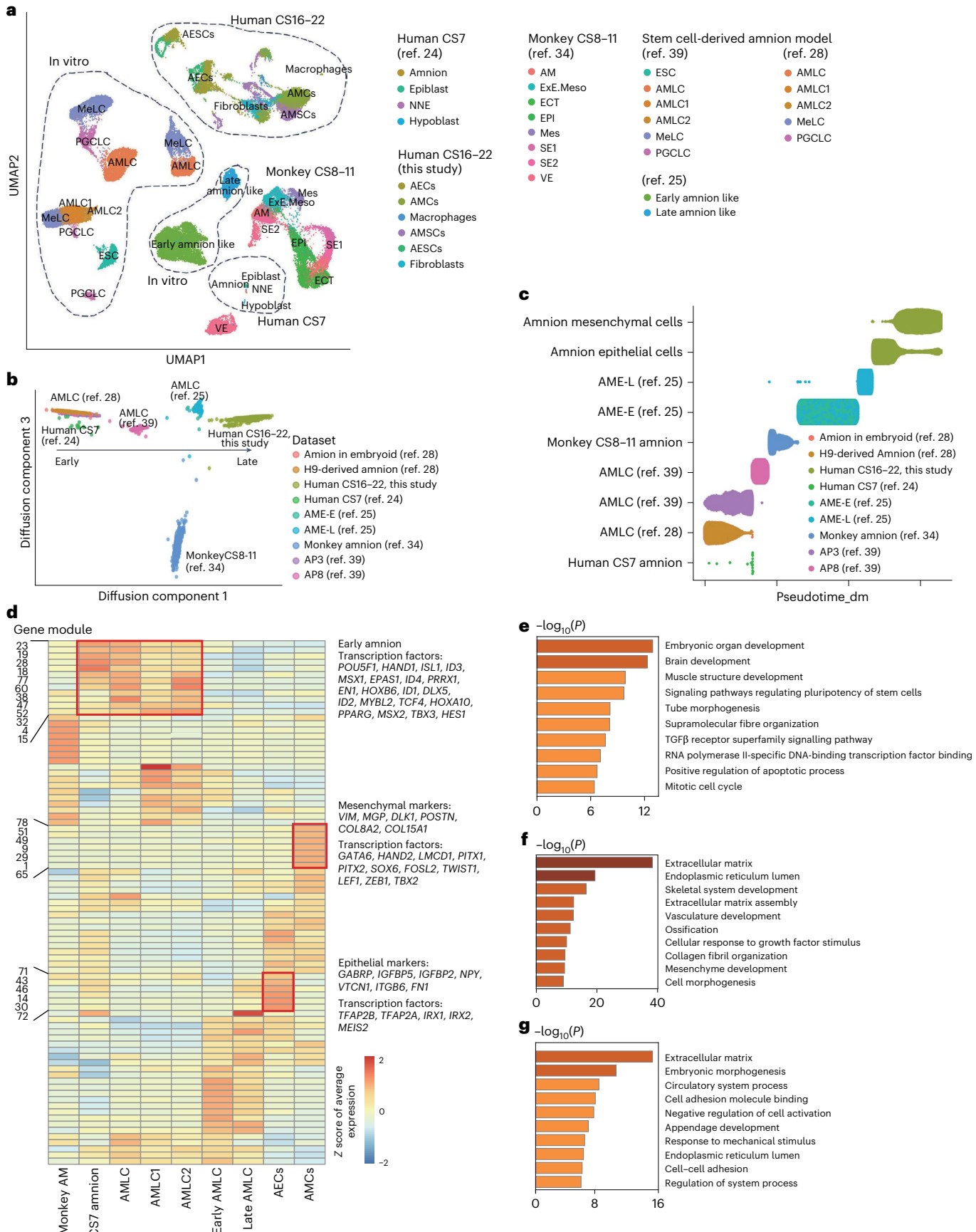

**Fig. 5 | Combined analysis of amnion data from human, monkey and in vitro stem cell-derived embryo models. a**, Combined UMAP of amnion data from human, monkey and in vitro stem cell-derived models. NNE, non-neural ectoderm; AM, amnion; ExE.Meso, extra-embryonic mesoderm; ECT, ectoderm; EPI, epiblast; Mes, mesenchyme; SE1, surface ectoderm1; SE2, surface ectoderm2; VE, visceral endoderm; ESC, embryonic stem cell; AMLC, amnion-like cell; MeLC, mesoderm-like cell; PGCLC, primordial germ cell-like cell. Data were merged from the scRNA-seq data generated in this study and five published datasets. **b**, A diffusion map illustrating the distribution of various amnion and amnion-like cell populations based on the integrated dataset comprising our data and five published datasets. Arrows indicate the developmental process. AME-E, amnion early like cell; AME-L, amnion late-like cell; AP3, hPS cells that were primed for 3 days; AP8, hPS cells that were primed for 8 days. **c**, Pseudotime plots of amnion and amnion-like cells derived from this study and five published datasets, with dataset origins indicated on the side. Pseudotime_dm; pseudotime computed using diffusion maps (dm). **d**, A heatmap showing the expression of each gene module across amnion and amnion-like cells from our data and five published datasets. Gene module numbers are shown on the left. Specific markers and transcription factors (TFs) are shown on the right. **e**–**g**, GO enrichment analysis of early amnion (**e**), AMCs (**f**) and AECs (**g**). Data sources include previously published datasets from human CS7 (ref. 24), monkey CS8–11 (ref. 34), and three sets of in vitro-derived amnion-like cells[25,28,39].

Although traditionally considered a non-neuronal tissue[45–47], the amnion has been reported to contain mesenchymal cells with neural progenitor-like characteristic[48–50] and neurotransmitter metabolism capabilities[51–53]. Furthermore, neural-related genes such as *SOX9*, *ID4* and *STMN2* have been detected in human amnion samples from both the first and second trimester[33]. Our results further show that SOX2-positive ectodermal cells also express neural markers such as *TUBB3* and *NR2F1*, suggesting that these neural progenitor-like cells in the amnion are probably ectodermal cells. However, higher-resolution immunofluorescence imaging would be required to confirm the presence of SOX2-positive cells and to refine our understanding of their morphology and spatial arrangement.

In our study, the entire amnion was collected without specific positional selection, and potential regional variations within the tissue were not specifically addressed. This limitation may contribute to the observed variability. Future studies using spatial transcriptomics or other positional mapping techniques could provide a deeper insight into the spatial heterogeneity and regional distinctions within the amnion.

Our study further demonstrates the potential developmental pathways of epithelial, mesenchymal and macrophage lineages in the human amnion. The observed EMT and epithelial-to-immune transitions (EIT) suggest dynamic cellular remodelling and immune modulation as the amnion develops during the first trimester of human pregnancy. These findings align with previous studies indicating the presence of EMT and EIT in the amnion and their importance in amniotic membrane remodelling[11,13,30,54]. However, further research will be necessary to provide deeper insights into these processes and their functional implications.

Beyond working as a protective barrier, the amnion is a major source of several growth factors crucial for embryogenesis, including EGF, FGF, PDGF and VEGF[55,56], which are involved in angiogenesis, tissue repair and immunomodulation. However, the specific cell types that secrete these growth factors remained unclear. Leveraging scRNA-seq data, we identified three distinct secretion patterns within the amnion: epithelial (MIF, WNT, BMP, GDF, PDGF and activin), mesenchymal (MDK, ncWNT, HGF, IGF and IL6) and macrophage (SPP1, TGF-β and CCL). Previous studies have shown that BMP4 promotes amnion development[28,39], and our recent work confirmed that BMP4 is essential for the epiblast differentiation into amnion in a stem cell-derived human embryo model[40]. Consistent with these results, we show here that BMP4 treatment of iPS cells led to the upregulation of both early and late amnion marker expression. Our findings reveal that immune cells within the amnion express TGF-β, a cytokine known to promote EMT[11]. This result suggests a potential mechanism in which macrophages may facilitate EMT via TGF-β secretion, thereby enhancing tissue repair[12]. Moreover, we observed that amnion cells express immunosuppressive factors such as SPP1[41], MIF[42–44] and TGF-β[57,58], which are known to inhibit immune responses. These findings further support the immunosuppressive properties of the amnion when used as a clinical biomaterial, highlighting its potential to modulate the immune environment and reduce inflammation during tissue repair. Clinically, this understanding can be leveraged to enhance wound healing, reduce

inflammation and promote tissue regeneration in applications such as treating burns, chronic wounds and organ injuries, demonstrating the amnion's promise in regenerative medicine.

Our comparative analysis of amnion RNA-seq data from human, non-human primate and in vitro amnion models indicates that stem cell-derived amnion models correspond to various stages of in vivo amnion development. Specifically, the amnion models derived from hPS cells using a microfluidic device and 3D biomimetic culture[28,39] appear to reflect earlier developmental stages compared with amnion-like cells derived from 2D culturing of hPS cells[25]. This difference might be linked to their respective culturing methods: microfluidic and 3D biomimetic culture systems more closely replicate the complex, dynamic conditions of the embryonic environment by providing a 3D, fluid-based context that supports cell–cell and cell–matrix interactions. This supports more accurate tissue development and spatial organization, resembling early stages of amnion development. By contrast, 2D culture systems tend to mimic a more mature structure of the amnion membrane by promoting flattened, layered cell growth, which may better reflect the architecture of foetal membranes. In addition, our study identified *TFAP2A* and *TFAP2B* as key transcription factors enriched in the development of AECs, while *GATA6*, *HAND2* and *SOX6* were highly expressed in AMCs. The expression patterns of these transcription factors suggest their involvement in regulatory pathways governing amnion development. However, their precise roles remain to be explored in future studies.

In conclusion, our findings provide comprehensive insights into the complex cellular architecture of amnion, highlighting its potential roles in embryonic development and tissue repair. This cellular map serves as a valuable resource for future functional studies on amnion development, in vitro amnion models and potential therapeutic applications.

## Online content

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

## Methods

### Human amnion collection

Human amnion tissue samples were collected from healthy pregnant donors after obtaining informed consent and following institutional ethical guidelines. All procedures were approved by the MRC-Wellcome Trust Human Developmental Biology Resource (HDBR) under ethical approval from the London – Fulham Research Ethics Committee (reference: 08/H0712/34+5, IRAS Project ID: 134561). Sample collection followed HDBR standard operating procedures and documentation, including: Patient Information Sheet and Consent Form, version 16; SOP – Recruitment of Donors, version 8; SOP – Collection of Consented Material, version 7; HDBR Background and Protocol, version 10. Tissue samples were obtained from elective caesarean sections or vaginal deliveries, with no known maternal or foetal complications. Detailed covariate information such as age, genotype or medical history of the donors was not available. When the amnion was collected, the yolk sac was readily identifiable as a distinct vascular sac, separate from the embryo, and could be separated from the amnion. The entire amnion was collected without any specific positional preference.

### Tissue processing

All tissues for sequencing were collected in HypoThermosol FRS preservation solution (H4416-100ML Merck) and stored at 4 °C until processing. Tissue dissociation was conducted within 24 h of tissue retrieval.

Tissues were cut into segments of less than 1 mm³ and washed with RPMI 10% FBS 1% penicillin–streptomycin medium before being digested with trypsin–EDTA 0.25% phenol red (25200072, Thermo Fisher Scientific) for 10–15 min at 37 °C with intermittent shaking. The digested tissue was passed through a 100-µm filter and the cells were collected by centrifugation (500g for 5 min at 4 °C). Cells were washed with PBS and resuspended in PBS 0.04% BSA before cell counting. In the case of the CS22 amnion sample, after digestion and washing, a reddish cell pellet was observed and red blood cell lysis buffer (eBioscience, 00-4333-57) was used for optimal lysis of erythrocytes in the single-cell suspension.

### 10x Genomics Chromium GEX (gene expression) library preparation and sequencing

For the scRNA-seq experiments, cells were loaded according to the manufacturer's protocol for the Chromium Next GEM Single Cell 5 v2 (dual index) kit for the CS17 amnion and Chromium Next GEM Single Cell 3 v3.1 (dual index) kit for the CS16, CS19 and CS22 amnion from 10x Genomics to attain 7,000 cells per reaction. Library preparation was carried out according to the manufacturer's protocol. Libraries were sequenced, aiming at a minimum coverage of 20,000 raw reads per cell, on the Illumina HiSeq4000 or Novaseq 6000 systems using the following sequencing format: read 1, 26 cycles; i7 index, 8 cycles, i5 index, 0 cycles; read 2, 98 cycles.

### 10x Genomics data preprocessing

Cell Ranger software from 10x Genomics was used for data preprocessing. Raw sequencing data were organized, with the requirement that sequencing reads be demultiplexed into FASTQ format files for each sample. The tool 'cellranger mkfastq' was used to demultiplex raw base call (BCL) files generated by Illumina sequencers into FASTQ files. Reads were mapped to the human reference genome hg38 and counted with GRCh38-3.0.0 annotation using 'cellranger count'. The data preprocessing workflow was streamlined and standardized to maintain consistency across samples.

### Quality control

Doublets and maternal cells were removed by the Souporcell software, using the default parameter with '--clusters=4'; only 'singlet' cells were kept for analysis. Souporcell clusters were shown in the UMAP; cells from a single genotype that clustered into the same Seurat cluster were identified as maternal cells and excluded from the analysis.

To eliminate contamination, we used the 'AddModuleScore' function in Seurat4 package to assign scores to cells identified as erythrocytes (markers: *HBZ, HBE1, HBG2, HBG1, HBA1, HBA2, HBM, ALAS2, HBB, GYPB, GYPC* and *GYPA*), chorion (markers: *CGA, CGB3, GCM1, CGB5, CGB7* and *CGB8*), yolk sac (markers: *AFP, CER1, HHEX, FOXA2* and *SPINK1*) and blood vessels (markers: *CD34, PECAM1, CLDN5, CDH5, ESAM, FLT1* and *OGN*). Only cells with scores <0 were kept for analysis.

We processed scRNA-seq data using a Seurat4. Initially, each dataset underwent quality control, filtering out cells on the basis of gene expression metrics, specifically retaining cells with gene counts (nFeature_RNA) between 500 and 8,000 and mitochondrial gene content (percent.mt) below 20%.

### Data integration

We normalized the data using the 'SCTransform' method from the Seurat package, using 'glmGamPoi' for normalization while regressing out the mitochondrial gene content and cell cycle genes. To mitigate batch effects and integrate data across different sources, we selected 5,000 integration features using Seurat's 'SelectIntegrationFeatures' function. 'PrepSCTIntegration' function prepared the datasets for integration, and 'FindIntegrationAnchors' was used to identify integration anchors, using SCT normalization. 'IntegrateData' function, with SCT normalization, was used to integrate the datasets.

### Dimension reduction and clustering

Dimensionality reduction and clustering were performed on the integrated scRNA-seq dataset. Principal component analysis was applied using the 'RunPCA' function. We used the 'RunUMAP' function to generate a UMAP representation, using the first 40 principal components (dims = 1:40) and specifying 'umap-learn' as the method. We constructed a shared nearest-neighbour graph using 'FindNeighbors', again focusing on the first 40 dimensions. The 'FindClusters' function was applied with a resolution parameter set to 0.4 to detect distinct cell clusters within the data.

### Identification of cluster-specific marker genes and cell type annotation

We normalized the data using the 'NormalizeData' function and proceeded to scale the data across all genes using the 'ScaleData' function. To identify markers for each cluster, we used the 'FindAllMarkers' function, focusing only on positive markers (only.pos = TRUE), and setting the minimum percentage of cells expressing the gene (min. pct) at 0.1 and the log fold change threshold (logfc.threshold) at 0.25. The resulting markers were grouped by their respective clusters and sorted to highlight the top markers based on average log fold change.

### GO enrichment

The Metascape (http://metascape.org) was utilized for comprehensive gene list annotation and enrichment analysis. GO enrichment analysis was performed across three main categories: biological process (BP), cellular component (CC) and molecular function (MF). For enrichment, we set 'Min Overlap' = 3, 'P Value Cutoff' = 0.01 and 'Min Enrichment' = 1.5.

### Trajectory and pseudotime analysis

Trajectory and pseudotime analysis was performed by Monocle3[36]. In brief, a CellDataSet (cds) was prepared from a Seurat object. To integrate UMAP coordinates from Seurat into Monocle3, we extracted the UMAP embedding from cds and aligned it with the Seurat UMAP embedding. We initiated trajectory analysis with 'learn_graph' and ordered cells based on UMAP trajectories using 'order_cells'. For identifying genes associated with cell trajectories, we used 'graph_test' on the cds,

specifying the 'neighbor_graph' as 'principal_graph' to highlight genes strongly related to the developmental pathways.

### RNA velocity

The BAM file was generated using the default parameters of Cell Ranger. For molecule counting, GRCh38 genome annotations from Cell Ranger's prebuilt references were utilized, categorizing the molecules into 'spliced', 'unspliced' and 'ambiguous'. Repeat annotation files were acquired from the UCSC Genome Browser. AnnData objects (h5ad) were built with ScanPy and ScVelo[35]. RNA velocity analysis was performed by the scVelo package in Python. In brief, we filtered and normalized the data using 'scv.pp.filter_and_normalize', setting a minimum threshold of 30 shared counts and selecting the top 2,000 highly variable genes to focus on the most informative aspects of the dataset. We calculated the neighbourhood moments with 'scv.pp.moments', using 30 principal components and 30 nearest neighbours, to capture the local structure and variability within the data. We estimated the RNA velocity using the stochastic model through 'scv.tl.velocity'. We constructed the velocity graph using 'scv.tl.velocity_graph' to visualize.

### Cell–cell communication

We used the CellChat[38] R package to analyse cell–cell communication networks based on the scRNA-seq dataset. A CellChat object was created by loading the normalized gene expression data into the CellChat environment, followed by assigning cell identities based on their respective metadata. We used the human ligand–receptor interaction database (CellChatDB.human) for our analysis, focusing specifically on 'Secreted Signaling' pathways to tailor our investigation towards secretome-mediated interactions. We computed the communication probabilities to infer the cellular communication networks. Communications between cell groups with fewer than ten cells were filtered out.

### Diffusion map

Diffusion map analysis was performed using the Destiny package[37], considering the 2,000 most variable protein-coding genes. 'Diffusion map 1' was set as pseudotime.

### Human, monkey and in vitro model data comparison

Human CS7 embryo data were downloaded from Array Express (E-MTAB-9388). Cells annotated as epiblast, amnion, hypoblast and non-neural ectoderm were selected for the analysis. Monkey CS8_CS11 data were obtained from Gene Expression Omnibus (GEO) under accession number GSE193007. Cells annotated as amnion, ectoderm, epiblast, extra-embryonic mesoderm, mesenchyme, surface ectoderm1, surface ectoderm2 and visceral endoderm were selected for the analysis. Stem cell-derived model data were downloaded from GSE179309, GSE205611 and GSE134571. Those datasets were integrated by the Seurat function 'merge'.

### Immunostaining

Human amnion sections were fixed using 4% paraformaldehyde and permeabilized with 0.1% Triton X-100. Non-specific binding was blocked using 10% donkey serum. The sections were incubated with primary antibodies (Supplementary Table 5), followed by washing and incubation with fluorophore-conjugated secondary antibodies. After additional washes, nuclear counterstaining was performed using DAPI. The samples were mounted with anti-fade medium and visualized under a confocal microscope.

### Human iPS cell culture

The human iPS cell line (WTC-11) was generously provided by Dr Bruce R. Conklin (Gladstone Institute of Cardiovascular Disease, UCSF). Human naive iPS cells were cultured on Matrigel-coated plates in 4CL medium[59] (1:1 mix of Neurobasal medium (Gibco, 21103049) and Advanced DMEM/F12 (Gibco, 12634028) supplemented with N2 (Gibco, 17502048) and B27 (Gibco, 17504044), sodium pyruvate (Corning, 25000CL), non-essential amino acids (Corning, 25025CL), GlutaMAX (Gibco, 35050061), penicillin–streptomycin (HyClone, SV30010), 10 nM DZNep (Selleck, S7120), 5 nM TSA (Vetec, V900931), 1 μM PD0325901 (Axon, 1408), 5 μM IWR-1 (Sigma, I0161), 20 ng ml$^{-1}$ human LIF (Peprotech, 300-05), 20 ng ml$^{-1}$ activin A (Peprotech, 120-14E), 50 μg ml$^{-1}$ L-ascorbic acid (Sigma, A8960) and 0.2% (v/v) Matrigel, Cells in 4CL were cultured at 37 °C, 5% $O_2$ and 5% $CO_2$. To induce specific signalling pathways, we supplemented the medium with 20 ng ml$^{-1}$ MDK (PeproTech, 450-16), 100 nM RA (STEMCELL Technologies, 72262) and 20 ng ml$^{-1}$ BMP4 (PeproTech, 120-05ET). These cells were maintained in culture and collected for analysis on the fifth day.

### Quantitative PCR (qPCR)

Total RNA was extracted using the Direct-zol RNA Purification Kit, Miniprep (Zymo Research, R2051), according to the manufacturer's protocol. RNA concentration and purity were assessed using a Nanodrop spectrophotometer. For qPCR, 10 ng of RNA per reaction was used with the Luna Universal One-Step RT-qPCR Kit (New England Biolabs, E3005L). Reactions were carried out under the following cycling conditions: reverse transcription at 55 °C for 10 min, initial denaturation at 95 °C for 1 min, followed by 40 cycles of 95 °C for 10 s and 60 °C for 30 s. Relative gene expression was calculated using the ΔΔCt method, with GAPDH as the internal control. Primers were synthesized by Integrated DNA Technologies (IDT). The sequences are presented in Supplementary Table 5.

### Statistics and reproducibility

Immunofluorescence staining experiments were repeated independently with consistent results. No statistical method was used to predetermine sample size. For scRNA-seq experiments, the number of samples was determined by tissue availability, which also guided collection and processing. Randomization was not applied during data collection or processing.

Putative maternal cells were excluded on the basis of the expression of maternal markers and genotype mismatches. This exclusion was performed before downstream analyses. Investigators were not blinded to group allocation during experiments or outcome assessment.

Statistical analyses were conducted using GraphPad Prism (v8.2.0). Data distribution was assumed to be normal, although this was not formally tested. For scRNA-seq analyses, differentially expressed marker genes were identified using Seurat (v4.3.0) with a two-sided Wilcoxon rank-sum test. Details regarding sample sizes, statistical tests and $P$ values are provided in the main text, figures, figure legends and supplementary tables.

### Reporting summary

Further information on research design is available in the Nature Portfolio Reporting Summary linked to this article.

## Data availability

Sequencing data that support the findings of this study have been deposited in the Gene Expression Omnibus (GEO) under accession number GSE260715. Previously published human CS7 embryo data were downloaded from Array Express (E-MTAB-9388). Monkey CS8_CS11 data were obtained from GEO under accession number GSE193007. Stem cell-derived model data were downloaded from GSE179309, GSE205611 and GSE134571. The human reference genome (GRCh38/hg38) used for alignment was downloaded from the 10x Genomics website (https://cf.10xgenomics.com/supp/cell-exp/refdata-gex-GRCh38-2020-A.tar.gz). Source data are provided with this paper. All other data supporting the findings of this study are available from the corresponding author on reasonable request. Source data are provided with this paper.

## References

59. Mazid, M. A. et al. Rolling back human pluripotent stem cells to an eight-cell embryo-like stage. *Nature* **605**, 315–324 (2022).

## Acknowledgements

We thank members of the Zernicka-Goetz lab and A. Andersen from Life Science Editors for their helpful comments; M. Shahbazi and B. Weatherbee for their assistance with accessing the amnion samples. This work was founded by Wellcome Trust grants (098287/Z/12/Z to M.Z.-G. and 220540/Z/20/A to R.V.-T.), NOMIS Foundation Award (M.Z.-G.) and Open Philanthropy grant (M.Z.-G.). C.W.G. was supported by the Leverhulme Trust.

## Author contributions

Samples were collected by N.S. and K.M. Data analyses were conducted by W.H., histology by C.W.G. and immunostaining by H.M.S. The project was conceptualized by M.Z.-G. The paper was written by W.H., with the help of M.Z.-G., R.V.-T. and C.W.G. Single-cell sequencing was performed by C.S.-S. and R.V.-T.

## Competing interests

The authors declare no competing interests.

## Additional information

**Extended data** is available for this paper at https://doi.org/10.1038/s41556-025-01696-9.

**Correspondence and requests for materials** should be addressed to Magdalena Zernicka-Goetz.

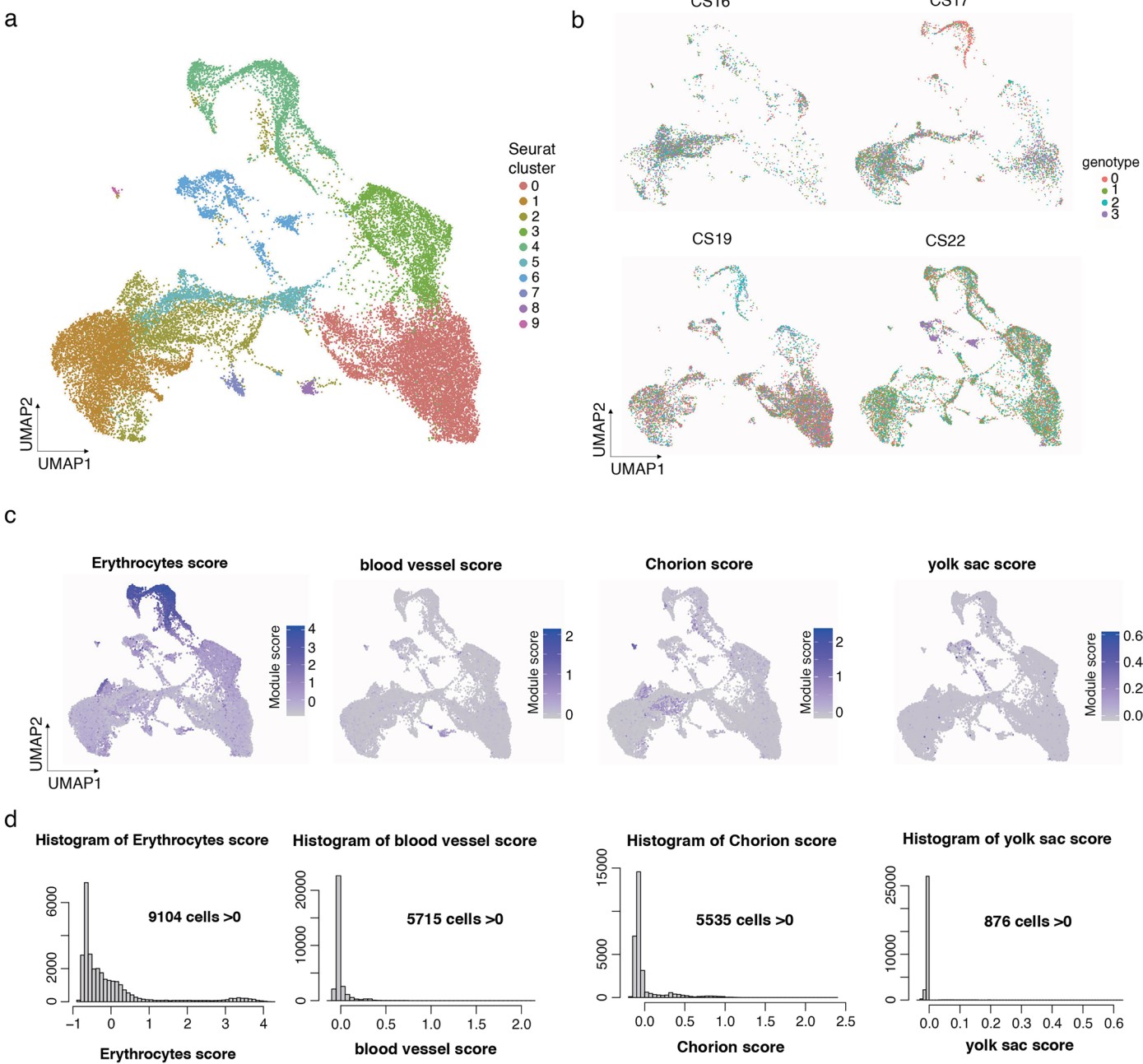

**Extended Data Fig. 1 | Quality control of single-cell RNA-seq data. (a)** UMAP visualization depicting Seurat-defined clusters using pre-QC data. (**b**) Overlay of Souporcell-defined genotypes on the pre-QC UMAP. Cells clustering by the same genotype were regarded as maternal cells. (**c**) Distribution of cell type scores on the pre-QC UMAP. (**d**) Histogram presenting the number of cells across different scoring categories. scRNA-seq analyses depicted in this figure are generated from human amnion samples of the following developmental stages: CS16 (n = 1), CS17 (n = 1), CS19 (n = 1), CS22 (n = 1).

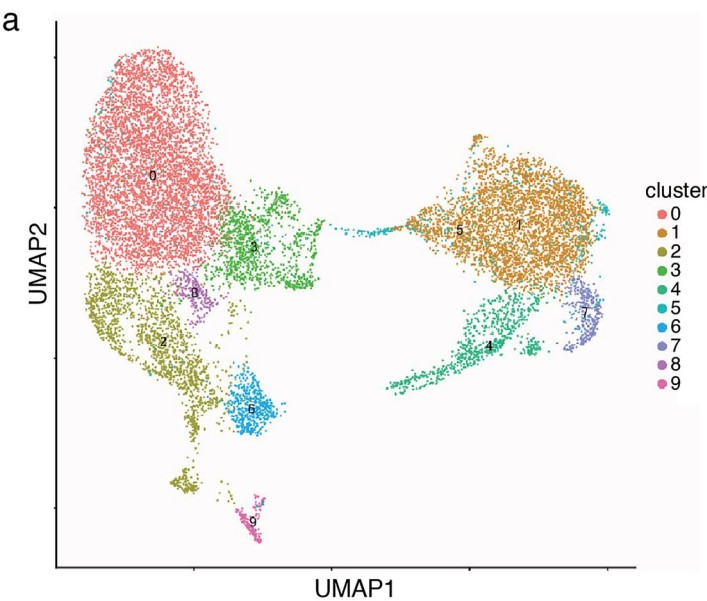

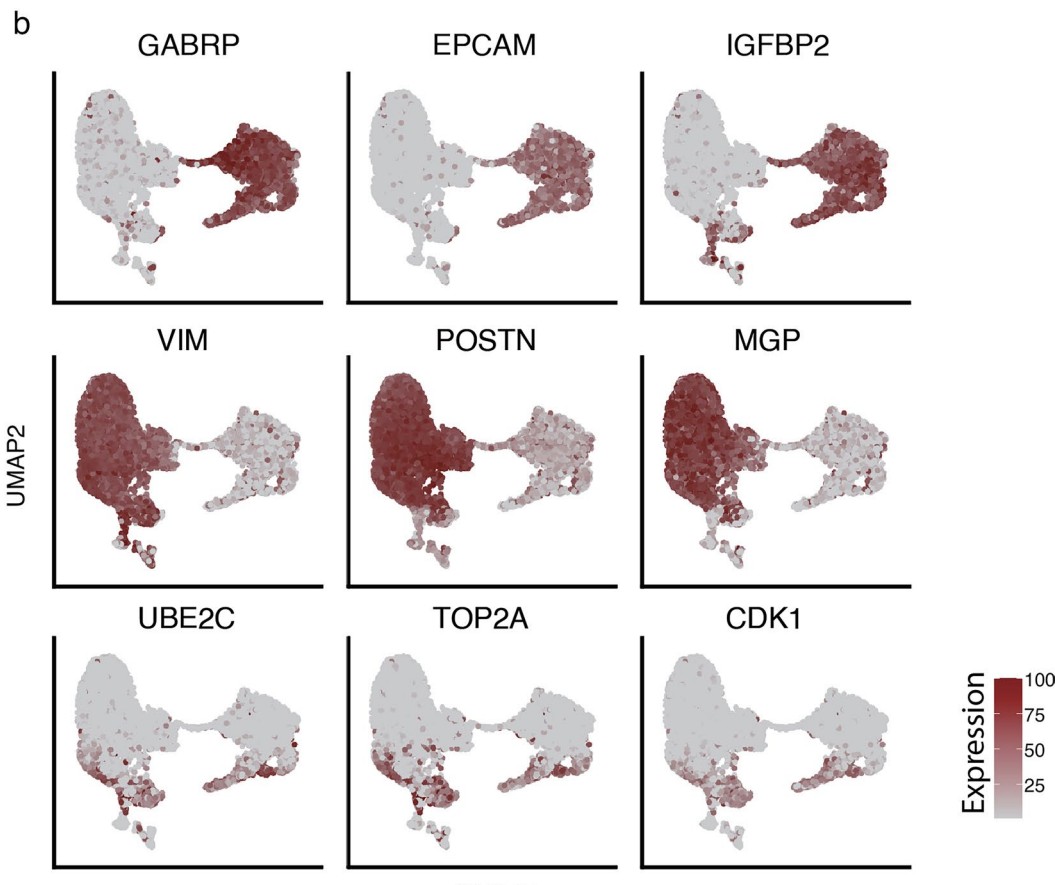

**Extended Data Fig. 2 | Seurat clusters and marker genes on UMAP. (a)** UMAP showing Seurat clusters after quality control **(b)** Marker genes of epithelial, mesenchymal, and progenitor cells were shown on UMAP. scRNA-seq analyses depicted in this figure are generated from human amnion samples of the following developmental stages: CS16 (n = 1), CS17 (n = 1), CS19 (n = 1), CS22 (n = 1).

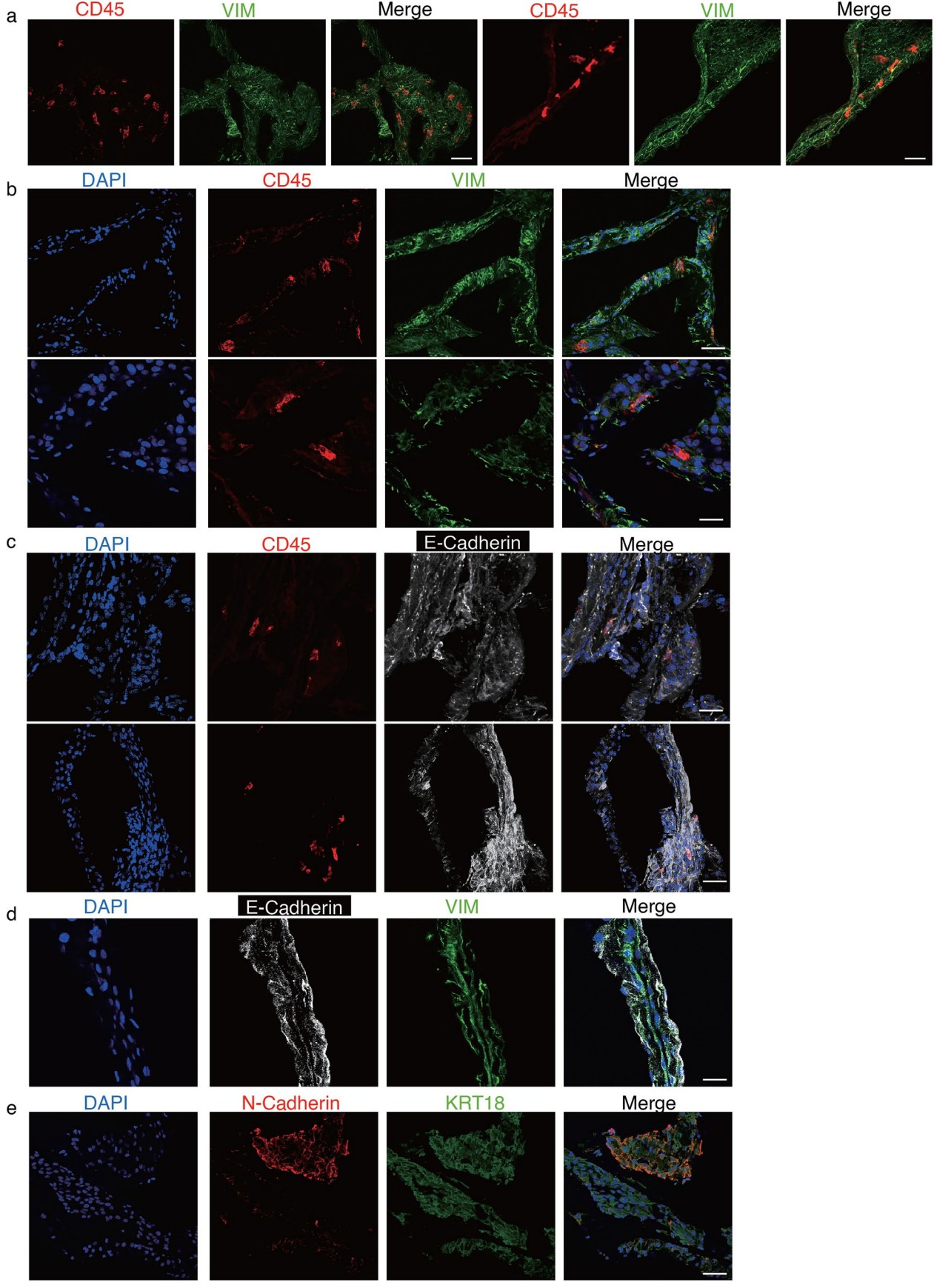

**Extended Data Fig. 3 | See next page for caption.**

**Extended Data Fig. 3 | Immunostainings of amnion sections in CS19 and CS23.** (**a**) Immunostaining of CD45 and VIM in the human amnion section. left panel: CS19; right panel: CS23. Scale bar = 50μm. Representative image from 2 independent experiments. (**b**) Immunostaining of CD45 and VIM in the human amnion section. top panel: CS19; bottom panel: CS23, Scale bar = 50μm. Representative image from 2 independent experiments. (**c**) Immunostaining of CD45 and E-Cadherin in the human amnion section. top panel: CS19; bottom panel: CS23, Scale bar=50μm. Representative image from 1 independent experiments. (**d**) Immunostaining of E-Cadherin and VIM in the human CS23 amnion section. Scale bar = 50μm. Representative image from 1 independent experiment. (**e**) Immunostaining of N-Cadherin and KRT18 in the human CS23 amnion section. Scale bar = 50μm. Representative image from 1 independent experiment.

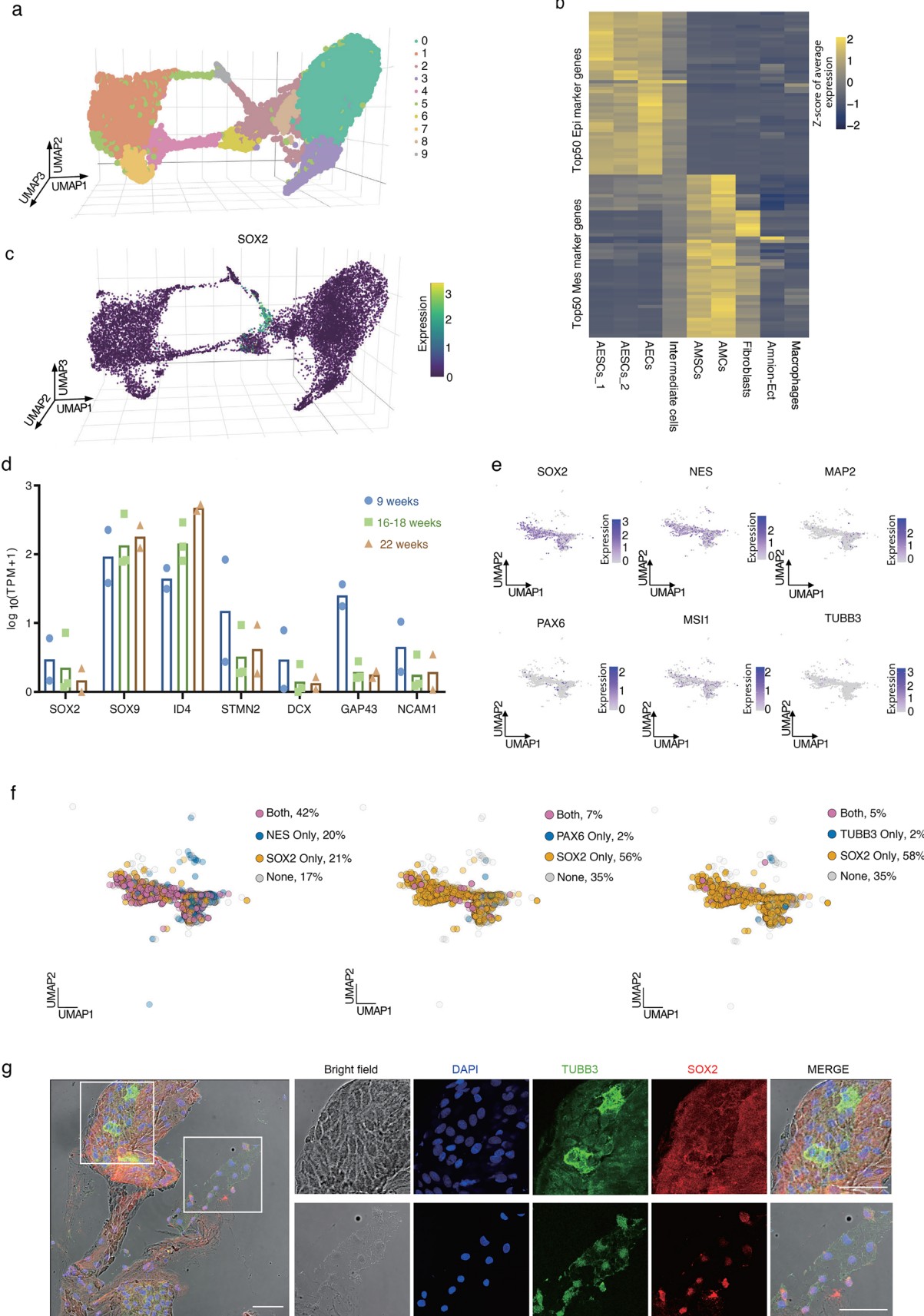

**Extended Data Fig. 4 | See next page for caption.**

**Extended Data Fig. 4 | Cell subtype identification in human amnion.**
(**a**) Seurat clusters on 3D UMAP. (**b**) Heatmap of the top 50 epithelial (top) and mesenchymal marker genes (bottom) across cell subtypes. (**c**) SOX2-positive cells were shown on UMAP. (**d**) Bar plot showing the expression of neural-related genes in 9, 16-18 and 22 weeks of human amnion samples (bulk data from Roost, 2015). The samples at 9 weeks and 22 weeks each have two biological replicates, while the samples from 16-18 weeks have three biological replicates. Each data point represents an individual sample, and bars indicate the mean expression value. (**e**) Umap showing the expression of neural-related genes in amnion cells. (**f**) Umap showing the Co-expression of SOX2 (yellow) and neural markers(blue).

Double-positive cells are colored in purple. Cell proportions are list on the legend. (**g**) Immunostaining of SOX2 and TUBB3 in the human CS19 amnion section. Scale bar=50μm. White frame region is magnified and shown on the right. Top panel shows TUBB3 positive cells, bottom panel shows TUBB3, SOX2 double-positive cells. Representative image from 4 independent experiments. scRNA-seq analyses depicted in (**a**), (**b**) and (**c**) are generated from human amnion samples of the following developmental stages: CS16 (n = 1), CS17 (n = 1), CS19 (n = 1), CS22 (n = 1). Analyses depicted in (**e**) and (**f**) are based on amnion cells from monkey CS8-11 dataset.

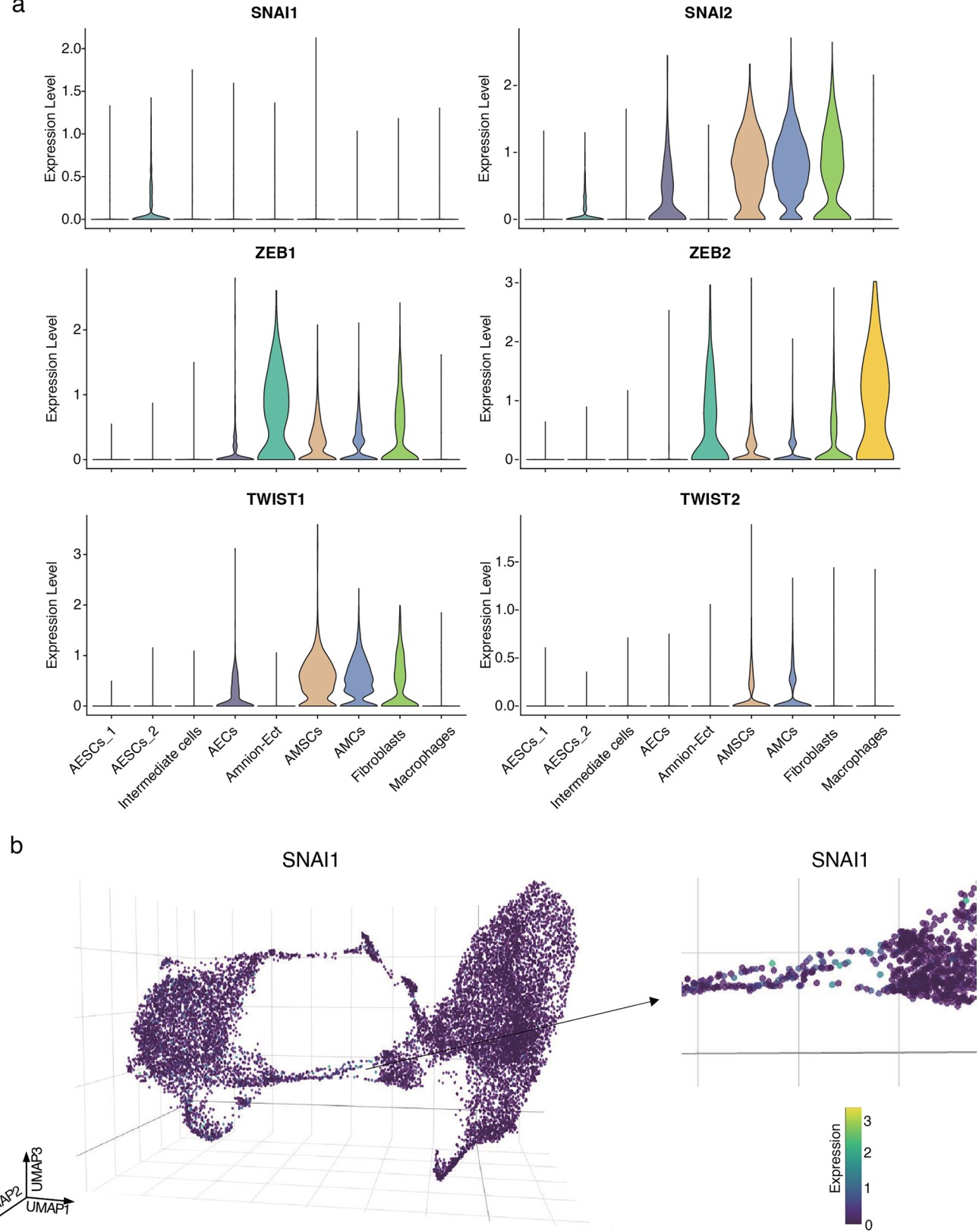

**Extended Data Fig. 5 | The expression of EMT-related TFs. (a)** Violin plots showing the expression of EMT-related TFs across different cell subtypes. **(b)** SNAI1-positive cells were shown on UMAP, with an enlarged section on the right. scRNA-seq analyses depicted in this figure are generated from human amnion samples of the following developmental stages: CS16 (n = 1), CS17 (n = 1), CS19 (n = 1), CS22 (n = 1).

a

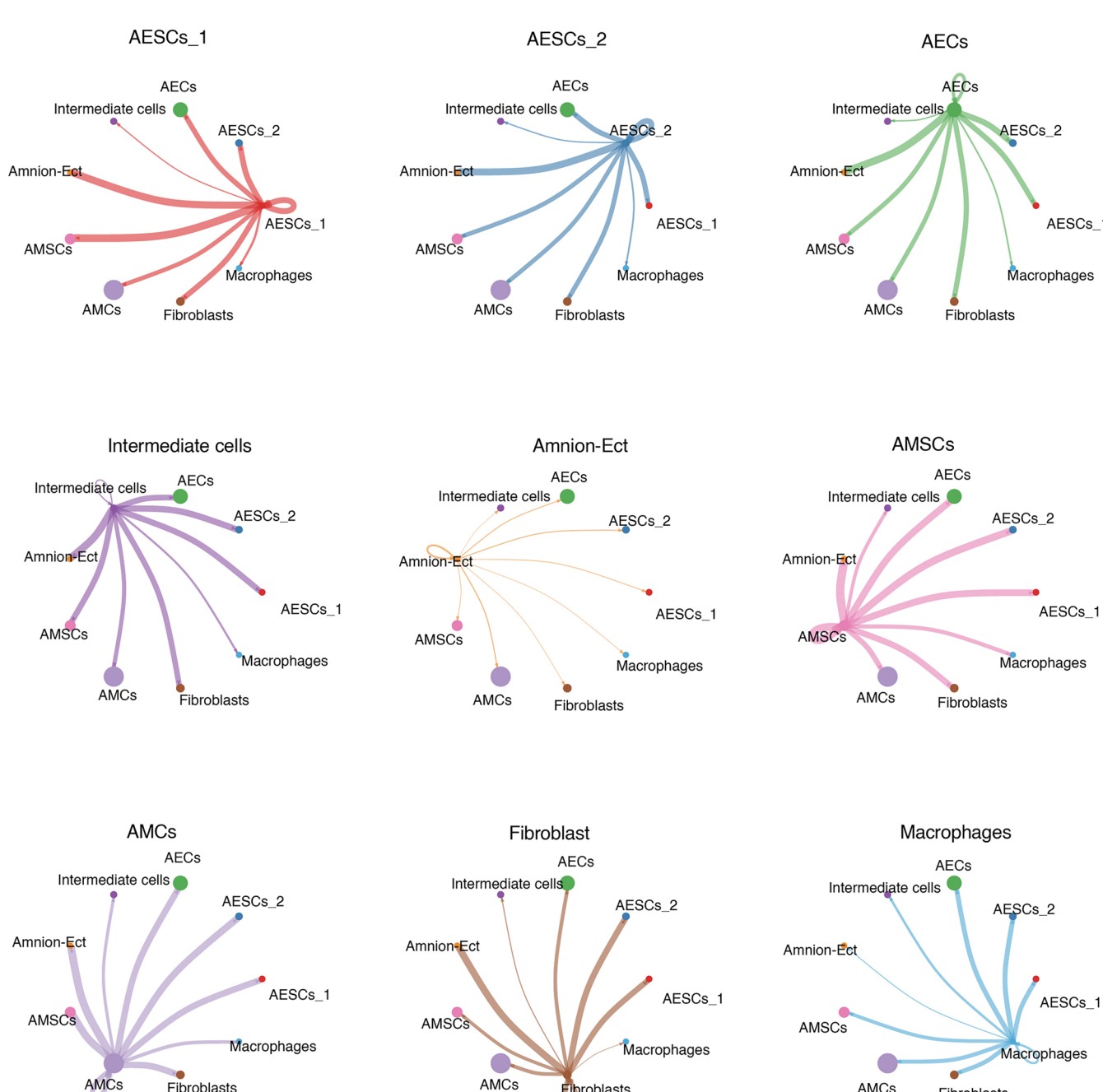

**Extended Data Fig. 6 | Cell-cell communication network between any two cell groups. (a)** Circle plots showing cell-cell interactions. Line weights indicating the total interaction strength. scRNA-seq analyses depicted in this figure are generated from human amnion samples of the following developmental stages: CS16 (n = 1), CS17 (n = 1), CS19 (n = 1), CS22 (n = 1).

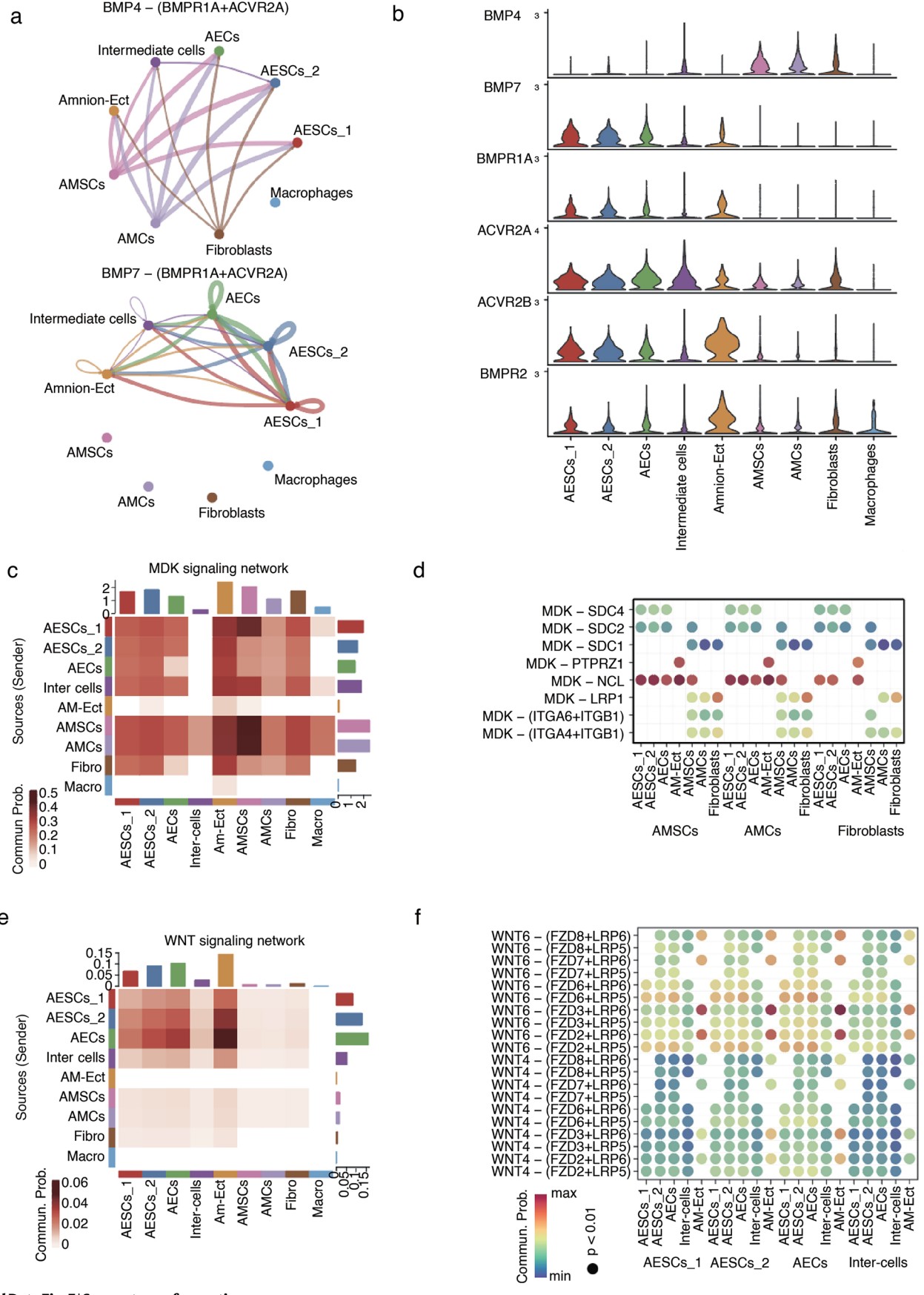

**Extended Data Fig. 7 | See next page for caption.**

**Extended Data Fig. 7 | Ligand/receptor interactions and gene expression.** (**a**) Circle plots depicting BMP ligand-receptor interactions. Upper panel: BMP4; Lower panel: BMP7. (**b**) Violin plots showing the expression of BMP ligands and their receptors across different cell types. (**c, e**) Heatmaps displaying the cell-cell interactions in the MDK (**c**) and WNT (**e**) signaling pathways. (**d, f**) Bubble plots showing MDK (**d**) and WNT (**f**) ligand-receptor interactions across different cell types. *Inter-cells*, short for intermediate cells. scRNA-seq analyses depicted in this figure are generated from human amnion samples of the following developmental stages: CS16 (n = 1), CS17 (n = 1), CS19 (n = 1), CS22 (n = 1).

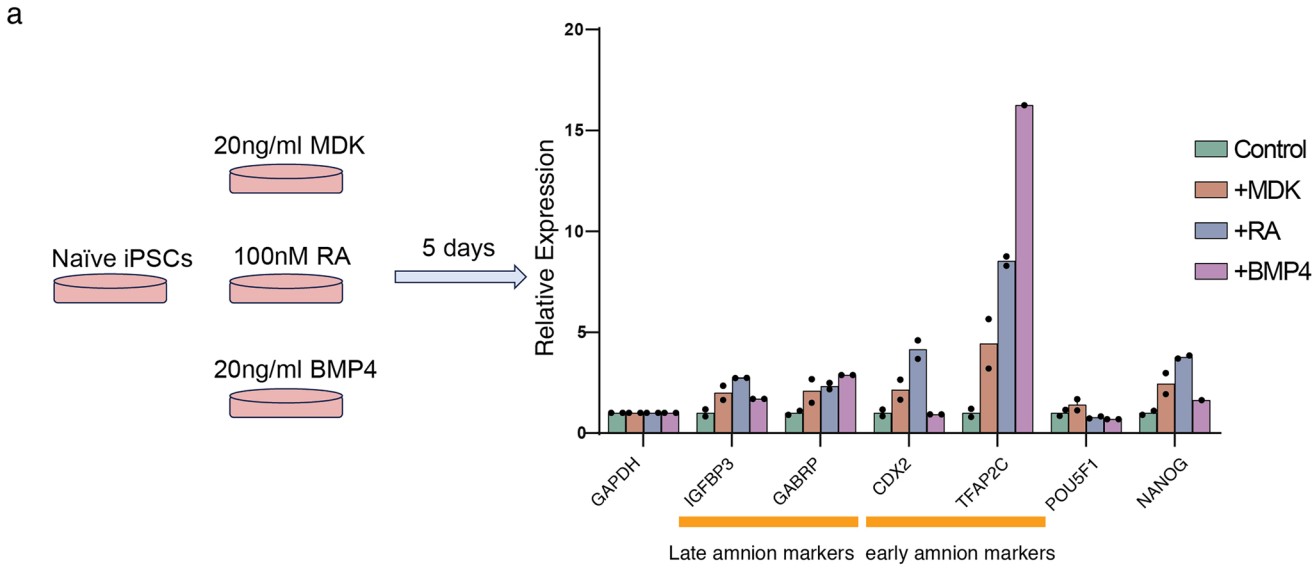

a

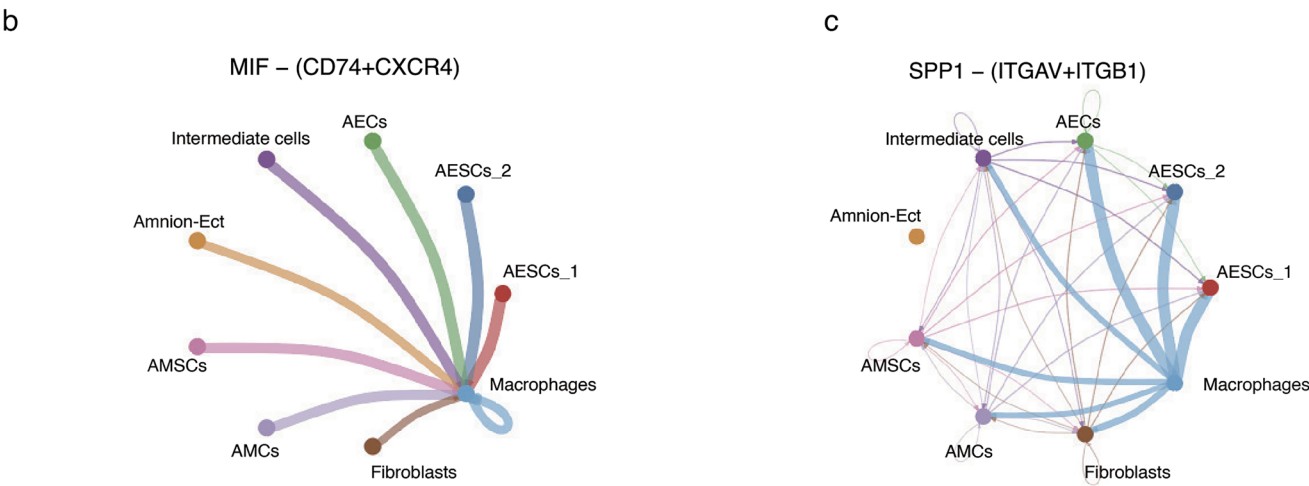

b

MIF – (CD74+CXCR4)

c

SPP1 – (ITGAV+ITGB1)

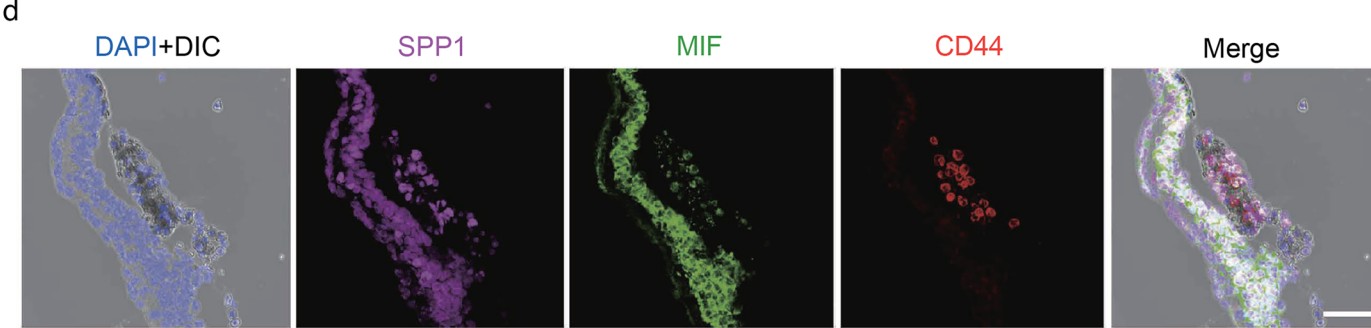

d

DAPI+DIC    SPP1    MIF    CD44    Merge

**Extended Data Fig. 8 | See next page for caption.**

**Extended Data Fig. 8 | BMP4 induction and Immunosuppressive factors expression in amnion.** (**a**) *In vitro* stem cell experiment design (left) and qPCR results (right) of cytokines-treated iPSCs. Relative gene expression was calculated using the ΔΔCt method, with GAPDH as the internal control. Each data point represents an individual sample, and bars indicate the mean expression value. The untreated medium was used as the control group, while groups treated with BMP4, RA, and MDK were the experimental groups. Each group, except for the BMP4-treated group, has two biological replicates. (**b, c**) Circle plots showing MIF (**b**) and SPP1 (**c**) interactions across different cell types. Analyses are generated from human amnion samples of the following developmental stages: CS16 (n = 1), CS17 (n = 1), CS19 (n = 1), CS22 (n = 1). (**d**) Immunostaining showing the expression of SPP1, MIF and CD44 in the CS16 amnion section, Scale bar = 50 μm. Representative image from 3 independent experiments.

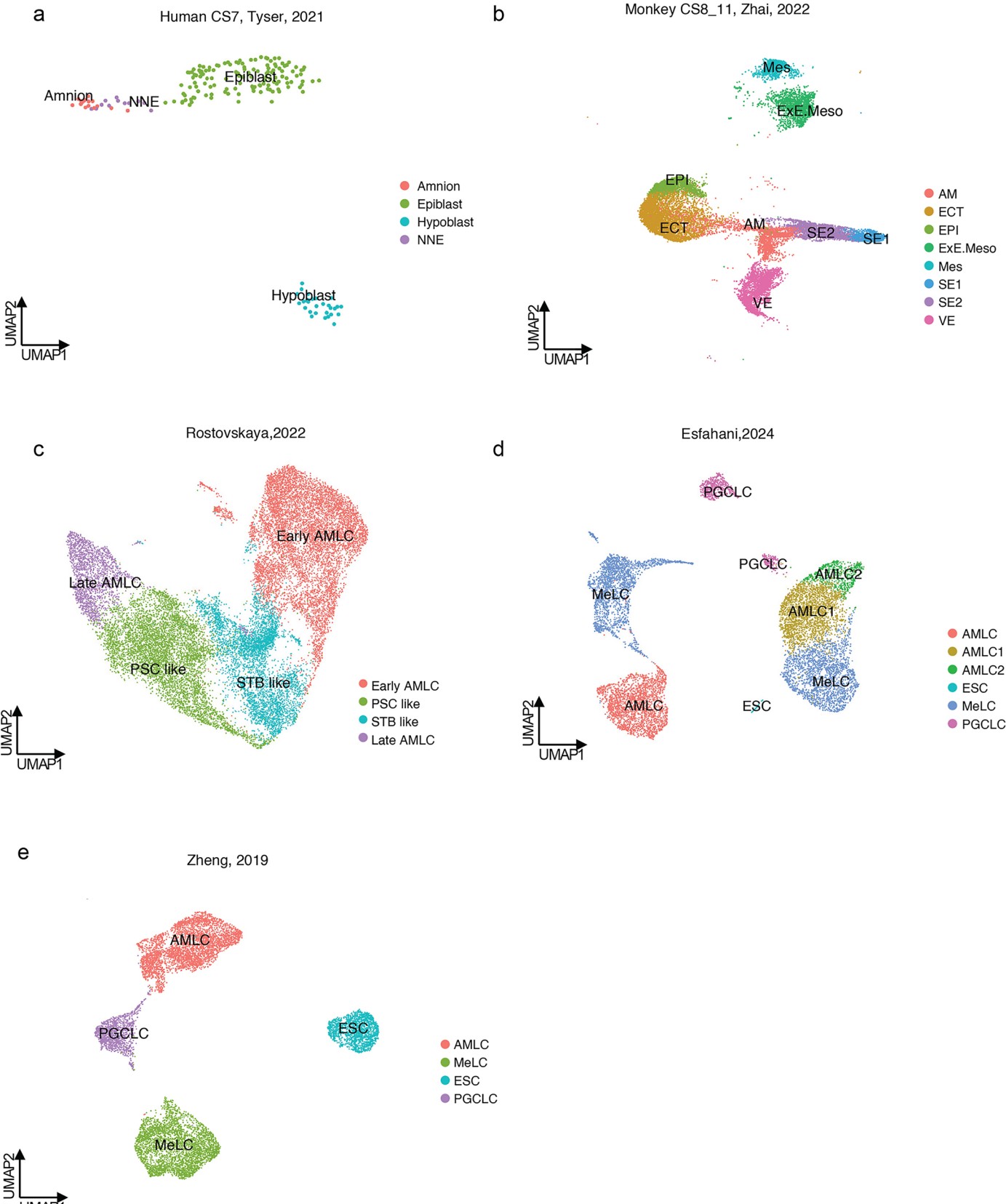

**Extended Data Fig. 9 | Single-cell RNA-seq datasets of amnion and amnion models.** (**a**) UMAP of single-cell RNA-seq data from the human CS7 amnion. (**b**) UMAP of single-cell RNA-seq data from the monkey CS8-11 amnion. (**c**–**e**) UMAPs of single-cell RNA-seq data from *in vitro* amnion models. Data were obtained from previously published studies (see Methods for details).

# Reporting Summary

## Statistics

For all statistical analyses, confirm that the following items are present in the figure legend, table legend, main text, or Methods section.

| n/a | Confirmed | |
|---|---|---|
| ☐ | ☒ | The exact sample size (*n*) for each experimental group/condition, given as a discrete number and unit of measurement |
| ☐ | ☒ | A statement on whether measurements were taken from distinct samples or whether the same sample was measured repeatedly |
| ☐ | ☒ | The statistical test(s) used AND whether they are one- or two-sided *Only common tests should be described solely by name; describe more complex techniques in the Methods section.* |
| ☒ | ☐ | A description of all covariates tested |
| ☐ | ☒ | A description of any assumptions or corrections, such as tests of normality and adjustment for multiple comparisons |
| ☐ | ☒ | A full description of the statistical parameters including central tendency (e.g. means) or other basic estimates (e.g. regression coefficient) AND variation (e.g. standard deviation) or associated estimates of uncertainty (e.g. confidence intervals) |
| ☐ | ☒ | For null hypothesis testing, the test statistic (e.g. *F*, *t*, *r*) with confidence intervals, effect sizes, degrees of freedom and *P* value noted *Give P values as exact values whenever suitable.* |
| ☒ | ☐ | For Bayesian analysis, information on the choice of priors and Markov chain Monte Carlo settings |
| ☒ | ☐ | For hierarchical and complex designs, identification of the appropriate level for tests and full reporting of outcomes |
| ☒ | ☐ | Estimates of effect sizes (e.g. Cohen's *d*, Pearson's *r*), indicating how they were calculated |

*Our web collection on statistics for biologists contains articles on many of the points above.*

## Software and code

Policy information about availability of computer code

| Data collection | For high-throughput sequencing data collection, we used Illumina HiSeq4000 or Nov aseq 6000 systems |
|---|---|
| Data analysis | For high throughput sequencing data processing and subsequent data analyses we used: linux packages: Cell Ranger(8.0.l), Souporcell( v2.5) R packages: R(4.2.2), Seurat(4.3.0), monocle3 (1.3.l),dplyr(l.l.2), CellChat(l.6.1), destiny(3.20) python packages: Python(3.ll.0), ScVelo(0.3.2), ScanPy(l.10.1), pandas(2.2.2), anndata( 0.10.7) Tools: Graph Prism(8.2.0), Metascape(v3.5) |

For manuscripts utilizing custom algorithms or software that are central to the research but not yet described in published literature, software must be made available to editors and reviewers. We strongly encourage code deposition in a community repository (e.g. GitHub). See the Nature Portfolio guidelines for submitting code & software for further information.

## Data

Policy information about <u>availability of data</u>

All manuscripts must include a <u>data availability statement</u>. This statement should provide the following information, where applicable:
- Accession codes, unique identifiers, or web links for publicly available datasets
- A description of any restrictions on data availability
- For clinical datasets or third party data, please ensure that the statement adheres to our <u>policy</u>

Sequencing data that support the findings of this study have been deposited in the Gene Expression Omnibus (GEO) under accession code GSE260715. Previously published human CS7 embryo data was downloaded from Array Express: E-MTAB-9388. Monkey CS8_CS11 data was obtained from GEO under accession number GSE193007. Stem cell derived model data were downloaded from GSE179309, GSE205611 and GSE134571. The human reference genome (GRCh38/hg38) used for alignment was downloaded from the 10x Genomics website (https://cf.10xgenomics.com/supp/cell-exp/refdata-gex-GRCh38-2020-A.tar.gz). Source data are provided with this study. All other data supporting the findings of this study are available from the corresponding author on reasonable request.

## Research involving human participants, their data, or biological material

Policy information about studies with <u>human participants or human data</u>. See also policy information about <u>sex, gender (identity/presentation), and sexual orientation</u> and <u>race, ethnicity and racism</u>.

| | |
|---|---|
| Reporting on sex and gender | We collected amnion tissue from embryos in the first trimester. Sex and genders are not identified in this study. |
| Reporting on race, ethnicity, or other socially relevant groupings | Race, ethnicity, or other socially relevant groupings are not identified in this study. |
| Population characteristics | We collected four human amnion samples representing weeks 5, 6, 7 and 8 of pregnancy and corresponding to Carnegie stages (CS) 16, 17, 19, and 22, respectively. Detailed covariate information such as age, genotype, or medical history of the donors was not available. All participants were healthy pregnant women who provided informed consent for the donation of amnion tissue. |
| Recruitment | Human amnion tissue samples were obtained through the MRC-Wellcome Trust Human Developmental Biology Resource (HDBR), from elective terminations of pregnancy with informed consent. Donors were not actively recruited by the research team. As samples were only obtained from individuals who consented to tissue donation for research purposes, potential self-selection bias may exist. Additionally, all samples were collected at collaborating clinical centers in the UK, which may limit generalizability to other populations. However, no information was available regarding the donors' demographic or clinical background beyond confirmation of healthy pregnancy status. |
| Ethics oversight | All procedures were approved by the MRC-Wellcome Trust Human Developmental Biology Resource (HDBR) under ethical approval from the London - Fulham Research Ethics Committee (Reference: 08/H0712/34+5, IRAS Project ID: 134561). Sample collection followed HDBR standard operating procedures and documentation, including: Patient Information Sheet and Consent Form, version 16; SOP – Recruitment of Donors, version 8; SOP – Collection of Consented Material, version 7; HDBR Background and Protocol, version 10. |

Note that full information on the approval of the study protocol must also be provided in the manuscript.

# Field-specific reporting

Please select the one below that is the best fit for your research. If you are not sure, read the appropriate sections before making your selection.

☒ Life sciences ☐ Behavioural & social sciences ☐ Ecological, evolutionary & environmental sciences

For a reference copy of the document with all sections, see <u>nature.com/documents/nr-reporting-summary-flat.pdf</u>

# Life sciences study design

All studies must disclose on these points even when the disclosure is negative.

| | |
|---|---|
| Sample size | No statistical methods were used to pre-determine sample size. Human amnion tissues were limited in availability, and all suitable samples from consenting donors were included in the study. Sample sizes are comparable to those reported in previous human single-cell studies. Across all samples, over 14,000 single cells were profiled, which provided sufficient resolution to identify cell populations and key gene expression patterns relevant to early development. |
| Data exclusions | To exclude potential maternal contamination, in our single-cell analysis, we carried out souporcell during data analysis. Cells from a single genotype that cluster into a Seurat cluster are identified as maternal cells and are excluded from the analysis. In addition, we used the "AddModuleScore" function in Seurat package to assign identify (score) and exclude: erythrocytes (markers: HBZ, HBEl, HBG2, HBGl, HBAl, HBA2, HBM, ALAS2, HBB, GYPB, GYPC, GYPA), chorion (markers: CGA, CGB3, GCMl, CGBS, CGB7, CGB8), yolk sac |

(markers: AFP, CERl, HHEX, FOXA2, SPIN Kl), and blood vessels (markers: CD34, PECAMl, CLDNS, CDHS, ESAM, FLTl, OGN). Only cells with scores <0 were kept for analysis

**Replication**

Replication was not applicable to single-cell RNA-seq samples as tissue availability from human donors was limited. All suitable samples obtained were included in the analysis, and scRNA-seq provides cellular-level resolution that captures biological variability across thousands of individual cells, partially mitigating the need for technical replication.

Immunofluorescence staining experiments were independently performed for multiple markers across different donor samples. Representative images are shown. 4 repeats were performed for CD45/VIM co-staining.2 repeats were performed for CD45/E-Cadherin co-staining. 1 repeat was performed for VIM/E-Cadherin and N-Cadherin/KRT18 staining. 2 repeats were performed for GABRP/SOX2 staining. 4 repeats were performed for E-Cadherin/TUBB3 staining. 3 repeats were performed for SPP1/CD44/MIF staining.

Stem cell experiments were independently repeated 2 times with consistent results.

**Randomization**

scRNA sequencing samples were not allocated into experimental groups. All available tissue samples from consenting donors were processed and analyzed as a single cohort. As this study is observational and exploratory in nature, group allocation and covariate control are not applicable.

Immunofluorescence (IF) staining was performed on available tissue samples from different human donors. Samples were not assigned to experimental groups, and no randomization or blinding was applied.

For stem cell experiments, samples were assigned to control and treatment groups based on experimental design. Allocation was randomized

**Blinding**

The investigators were not blinded to sample identity during data collection or analysis, as the study was exploratory in nature and did not involve treatment allocation or intervention groups. The investigators were not blinded to group allocation during the stem cell experiments or data analysis.

# Reporting for specific materials, systems and methods

We require information from authors about some types of materials, experimental systems and methods used in many studies. Here, indicate whether each material, system or method listed is relevant to your study. If you are not sure if a list item applies to your research, read the appropriate section before selecting a response.

## Materials & experimental systems

| n/a | Involved in the study |
|---|---|
| ☐ | ☒ Antibodies |
| ☐ | ☒ Eukaryotic cell lines |
| ☒ | ☐ Palaeontology and archaeology |
| ☒ | ☐ Animals and other organisms |
| ☒ | ☐ Clinical data |
| ☒ | ☐ Dual use research of concern |
| ☒ | ☐ Plants |

## Methods

| n/a | Involved in the study |
|---|---|
| ☒ | ☐ ChIP-seq |
| ☒ | ☐ Flow cytometry |
| ☒ | ☐ MRI-based neuroimaging |

## Antibodies

**Antibodies used**

GABRP Polyclonal Antibody, Thermo Fisher, Cat#PA5-46830
Vimentin (5G3F10), cell signal technology, mAb #3390
SOX2 Monoclonal Antibody (Btjce), Thermo Fisher,Catalog # 14-9811-82, clone: Btjce
Osteopontin Monoclonal Antibody (2F10), Thermo Fisher, Cat# 14-9096-82, clone: 2F10
MIF Polyclonal Antibody, Thermo Fisher, Cat# PA5-27343
CD44 Monoclonal Antibody (IM7) Thermo Fisher, Cat# 14-0441-82, clone: IM7
CD324 (E-Cadherin) Monoclonal Antibody (DECMA-1),Thermo Fisher,Catalog # 16-3249-82,RRID:AB_10734213, clone: DECMA-1
CD45 (Intracellular Domain) (D9M8I) , Cell signal technology, Cat#13917T, clone: D9M8I
Anti-N Cadherin antibody [5D5], Abcam, Cat#ab98952, RRID:AB_10696943, clone: 5D5
Anti-Keratin 18 antibody, Sigmaaldrich, Cat#SAB4501665, RRID:AB_10746153
Purified anti-Tubulin β 3 (TUBB3) Antibody, Biolegend, Cat#801213, RRID:AB_2313773, clone: TUJ1
Donkey anti-Rabbit IgG (H+L) Highly Cross-Adsorbed Secondary Antibody, Alexa Fluor™ 488 Thermo Fisher Cat# A-21206
Donkey anti-Mouse IgG (H+L) Highly Cross-Adsorbed Secondary Antibody, Alexa Fluor™ 568 Thermo Fisher Cat# A10037
Donkey anti-Rat IgG (H+L) Highly Cross-Adsorbed Secondary Antibody, Alexa Fluor™ Plus 647 Thermo Fisher Cat# A48272
Goat anti-Rat IgG (H+L) Cross-Adsorbed Secondary Antibody, Alexa Fluor™ 647 Thermo Fisher Cat# A-21247

**Validation**

Anti-GABRP (Thermo Fisher, Cat# PA5-46830) is validated by the manufacturer for human samples in IHC and WB applications.
Anti-Vimentin (clone 5G3F10, Cell Signaling Technology, Cat# 3390) is a mouse monoclonal antibody validated by the manufacturer for immunofluorescence and human samples. The antibody has been widely used in published studies involving human tissues and cell lines, and produces a clear cytoplasmic staining pattern consistent with known Vimentin expression.
Anti-SOX2 (clone Btjce, Thermo Fisher, Cat# 14-9811-82) is a mouse monoclonal antibody validated by the manufacturer for human samples in flow cytometry applications.
Anti-Osteopontin (clone 2F10, Thermo Fisher, Cat# 14-9096-82) is a mouse monoclonal antibody validated by the manufacturer for flow cytometry in human and mouse samples.
Anti-MIF (Thermo Fisher, Cat# PA5-27343) is a rabbit polyclonal antibody validated by the manufacturer for use in human tissues in

Western blot and immunohistochemistry applications.

Anti-CD44 (clone IM7, Thermo Fisher, Cat# 14-0441-82) is a rat monoclonal antibody validated by the manufacturer for use in human samples in flow cytometry.

Anti-E-Cadherin (CD324) (clone DECMA-1, Thermo Fisher, Cat# 16-3249-82, RRID: AB_10734213) is a mouse monoclonal antibody validated by the manufacturer for immunofluorescence and for use in human samples. The antibody showed membrane-localized staining consistent with known E-Cadherin expression in epithelial cells. It has been widely used in published studies involving human tissues.

Anti-CD45 (Intracellular Domain) (clone D9M8I, Cell Signaling Technology, Cat# 13917T, RRID: AB_2799583) is a rabbit monoclonal antibody validated by the manufacturer for immunofluorescence in human samples. In our study, it produced specific intracellular staining patterns consistent with known CD45 expression in hematopoietic cells.

Anti-N-Cadherin (clone 5D5, Abcam, Cat# ab98952, RRID: AB_10696943) is a mouse monoclonal antibody validated by the manufacturer for immunofluorescence and for use in human tissues. In our study, the antibody showed strong membrane staining consistent with known N-Cadherin localization in mesenchymal cell populations.

Anti-Keratin 18 (Sigma-Aldrich, Cat# SAB4501665, RRID: AB_10746153) is a rabbit polyclonal antibody validated by the manufacturer for immunofluorescence and for use in human samples. The antibody produced strong cytoplasmic staining consistent with known expression of Keratin 18 in epithelial cell types.

Anti-Tubulin β 3 (TUBB3) (clone TUJ1, BioLegend, Cat# 801213, RRID: AB_2313773) is a mouse monoclonal antibody validated by the manufacturer for immunofluorescence and for use in human samples. This antibody is widely used as a neuronal marker and produced strong cytoplasmic staining in TUBB3+ cell populations consistent with known expression patterns.

# Eukaryotic cell lines

Policy information about cell lines and Sex and Gender in Research

| Cell line source(s) | This iPSC line (WTC-11) generously provided by Dr. Bruce R. Conklin (Gladstone Institute of Cardiovascular Disease, UCSF) |
|---|---|
| Authentication | Induced pluripotent stem cell line derived from adult skin (leg) fibroblasts; subject clinically normal; wild-type; FISH test: 46, XY. This iPSC line (WTC-11) was produced by Dr. Bruce R. Conklin (Gladstone Institute of Cardiovascular Disease, UCSF) |
| Mycoplasma contamination | All cell lines were routinely tested every two weeks to ensure that they were not contaminated with Mycoplasma |
| Commonly misidentified lines (See ICLAC register) | No commonly misidentified cell lines listed by the International Cell Line Authentication Committee (ICLAC) were used in this study. |

# Plants

| Seed stocks | *Report on the source of all seed stocks or other plant material used. If applicable, state the seed stock centre and catalogue number. If plant specimens were collected from the field, describe the collection location, date and sampling procedures.* |
|---|---|
| Novel plant genotypes | *Describe the methods by which all novel plant genotypes were produced. This includes those generated by transgenic approaches, gene editing, chemical/radiation-based mutagenesis and hybridization. For transgenic lines, describe the transformation method, the number of independent lines analyzed and the generation upon which experiments were performed. For gene-edited lines, describe the editor used, the endogenous sequence targeted for editing, the targeting guide RNA sequence (if applicable) and how the editor was applied.* |
| Authentication | *Describe any authentication procedures for each seed stock used or novel genotype generated. Describe any experiments used to assess the effect of a mutation and, where applicable, how potential secondary effects (e.g. second site T-DNA insertions, mosiacism, off-target gene editing) were examined.* |

