## [Peer Review File · Nature Cell Biology]

Atlas of Amnion Development During the First Trimester of Human Pregnancy

Corresponding Author: Professor Magdalena Zernicka-Goetz

Version 0:

Decision Letter:

*Please delete the link to your author homepage if you wish to forward this email to co-authors.

Dear Magda,

Your manuscript, "Atlas of amnion development during the first trimester of human pregnancy", has now been seen by 3 referees, who are experts in human development, atlases (referee 1); human development, scRNAseq, atlas, amnion (referee 2); and human development, amnion, single cell transcriptomics (referee 3). As you will see from their comments (attached below) they find this work of potential interest, but have raised very important concerns, which in our view would need to be thoroughly addressed with considerable revisions before we can consider publication in Nature Cell Biology.

Nature Cell Biology editors discuss the referee reports in detail within the editorial team, including the chief editor, to identify key referee points that should be addressed with priority, and requests that are overruled as being beyond the scope of the current study. To guide the scope of the revisions, I have listed these points below.

*** I should stress that the referees' concerns point to a premature dataset and these points would need to be addressed with experiments and data, and reconsideration of the study for this journal and re-engagement of referees would depend on strength of these revisions. ***

In particular, it would be essential to:

(A) Provide experimental evidence to exclude contamination, as indicated by:

Reviewer #1:

"The amnion is connected to other tissues in early embryonic development. The authors should provide evidences to exclude the contamination of other tissues. In addition, different regions of amnion may exhibit different cell composition. The amnion samples are intact? The authors should clarify them".

"For the presence of ectodermal cells (neural stem cells or progenitors) in amnion, additional evidences, such as immunostaining of specific neural ectodermal markers, are critical because the potential contamination from neural ectodermal tissue in the fetus needs to be excluded".

"The authors observed the presence of blood and immune cells in the amnion. This result is surprising. Although the authors exclude maternal contamination, as well-known, yolk sac and placental villous tissue can produce blood and immune cells in early fetal stage. How to exclude the possible contamination of blood and immune cells produced by yolk sac and placental villous tissue?"

Reviewer #3:

"The amnion at 6-8WPC is a transparent layer containing no erythroblasts or erythroid or any type of vasculature and to my knowledge no smooth muscle cells or fibroblast or endothelial cells (there is no vasculature).

This is very different from the situation at term, as the avascular amnion hss fused with the vascularised chorion/placental plate. The cell types identified seem to come from term amnion and seem unrelated to the fetal state? It is quite puzzling that so many different cell types were observed in such young amnions. These cell types are not expected to be present in the 6-8WPC amnion as such. Hence, providing robust validation is crucial.

I am not sure why this is the case but I would advise extreme caution, as there may be issues with the purity of the samples used or perhaps yolk sac was included as well, as that tissue is vascularized and would contain fetal blood. It would therefore be very important to include images and or histology of the actual (or comparable) tissue used before it went for sequencing. Again, including more extensive IF is crucial to reassure and validate the cell types described, which are unexpected".

"It is good that the authors separated the cells by genotype to exclude that some of the cell types encountered are indeed from maternal origin, such as immune cells. However, I would have preferred to see this presented at the start of the work, when QC is discussed, and information about number and cell type of maternal cells (mostly maternal blood cells I assume) should be explicitly mentioned and stratification between the genotypes of embryo and mother should be provided. During the abortion procedure there is obvious extreme contamination with maternal blood and I am not sure how confident the authors are that the material used for sequence was sufficiently washed from (maternal and fetal) blood and no yolk sac was present. For example, Fig5i seem to be contaminated with (maternal and fetal) blood. How has this been assessed and or prevented? The amnion is a smooth membrane without any blood cells attached, so

Fig5i suggests contamination with fetal or maternal blood".

(B) Provide as much of experimental validation of the sequencing results as possible, and strengthen current claims, as requested by:

Reviewer #1:

"If immunofluorescence staining could be provided for some important cell population, it is critical to help readers understand the cell subsets annotated by the scRNAseq data in Figure 1d".

Reviewer #2:

"The manuscript should more explicitly compare and contrast its findings with existing literature on amnion development and cellular composition (PMID: PMID: 35439430). This includes detailing how the identified cell types and signaling pathways align with or expand upon previous studies. Such a comparison would highlight the novel contributions of this research more clearly. Also including DATASET at earlier stages such as CS7 human datasets and also CS8-11 monkey ones could help to better illustrating amnion development".

Reviewer #3:

"I think it is possible that the so-called 'amnion progenitor' cluster could be a different population, located at specific place, for example, at the border with the embryo or close to the umbilical cord, that could simply have a mixed signature meso/ecto? The pseudotime analysis is interesting but with only 3 tissue donors of similar age range it may be inconclusive/premature to infer directionality (the change in directionality is interesting but should be more robustly determined). I suggest the inclusion of more individuals of the same age to exclude interdonor variation and making sure the amnion tissue used is from a similar area, as regional differences may exist and be confounding the data. Hence to annotate as 'progenitor' is premature".

"Direct comparison with the Tyser data that included the amnion (annotated as non-neural ectoderm & advanced mesoderm) should be provided to aid in the improvement of the cell type annotation of the Tyser data; and to generate a developmental trajectory. I would also request the integration (or at least a comparison) of the human data with comparable stages of amnion development from the monkey, as this data are well annotated and should be available for direct comparison and/or integration".

"It would be very important to perform robust validation using IF. The work includes two IF and more (all) cell types should be identified. One of the two AB used GATA3 is not present in the bioinformatics analysis, hence it is unclear why other markers were not used to validate the expected cell types in the amnion. The 6-8WPC embryo is a large membrane and can be used for many stainings. I am sure the labs involved have the ABs for many of the markers identified, such as SOX2, TUBB3 or panKRT".

(C) Analyze additional samples, as requested by:

Reviewer #1:

"The main weakness of this study lacks immunostaining evidences to confirm single-cell RNA-Seq results, and only one sample every stage also weak the conclusion of the manuscript".

"As the authors mentioned that each timepoint is represented by a single sample, the result in Fig. 1e is inconclusive and misleading. I recommend to remove this result".

Reviewer #3:

"The ages analyzed are very close together and it would be more appropriate to consider a wider spread of embryonic ages to generate a bonfide atlas, this seems more a single age group (6-8WPC), there is no separated analysis by sex and the n used is a bare minimum (n=3)".

(D) All other referee concerns pertaining to strengthening existing data, providing methodological details, clarifications and textual changes, should also be addressed.

(E) Finally, please pay close attention to our guidelines on statistical and methodological reporting (listed below) as failure to do so may delay the reconsideration of the revised manuscript. In particular please provide:

We would be happy to consider a revised manuscript that would satisfactorily address these points, unless a similar paper is published elsewhere, or is accepted for publication in Nature Cell Biology in the meantime.

In contrast, although we agree with referees #1 and #3 that addressing their specific points would provide valuable insights, we consider these points to be beyond the scope of the present study. Thus, addressing them experimentally will not be necessary for reconsideration of the manuscript at this journal:

* Rev #1:

"Bipotent progenitors could be further clustered into two subgroups represented by cluster 2 and cluster 3. Fig. 2e, f and Fig. S2h showed that AESC and AMSC come from the different two cluster, respectively, which imply that there may be no any bipotent progenitors that has the ability to give rise to both AESC and AMSC. More evidences are required to demonstrate the bipotent potential".

* Rev #2:

"While the identification of signaling pathways and cell types provides a framework for understanding amnion development, experimental validation of these functional roles is critical. Specifically, in vitro or in vivo experiments demonstrating the roles of identified signaling pathways (e.g., BMP4/BMPRII, PDGFA/PDGFR) in amnion development and function would greatly strengthen the manuscript's conclusions".

- ensure that it conforms to our format instructions and publication policies (see below and www.nature.com/nature/authors/).
- provide a point-by-point rebuttal to the full referee reports verbatim, as provided at the end of this letter.
- provide the completed Editorial Policy Checklist (found here <https://www.nature.com/authors/policies/Policy.pdf>), and Reporting Summary (found here <https://www.nature.com/authors/policies/ReportingSummary.pdf>). This is essential for reconsideration of the manuscript and these documents will be available to editors and referees in the event of peer review. For more information see <http://www.nature.com/authors/policies/availability.html> or contact me.

Nature Cell Biology is committed to improving transparency in authorship. As part of our efforts in this direction, we are now requesting that all authors identified as 'corresponding author' on published papers create and link their Open Researcher and Contributor Identifier (ORCID) with their account on the Manuscript Tracking System (MTS), prior to acceptance. ORCID helps the scientific community achieve unambiguous attribution of all scholarly contributions. You can create and link your ORCID from the home page of the MTS by clicking on 'Modify my Springer Nature account'. For more information please visit <http://www.springernature.com/orcid>.

Link Redacted

We would like to receive a revised submission within six months. We would be happy to consider a revision even after this timeframe, however if the resubmission deadline is missed and the paper is eventually published, the submission date will be the date when the revised manuscript was received.

We hope that you will find our referees' comments, and editorial guidance helpful. Please do not hesitate to contact me if there is anything you would like to discuss.

Best wishes,

Stelios

Stylios Lefkopoulos, PhD
He/him/his
Senior Editor, Nature Cell Biology
Springer Nature
Heidelberger Platz 3, 14197 Berlin, Germany

E-mail: stylios.lefkopoulos@springernature.com
Twitter: @s_lefkopoulos
LinkedIn: [linkedin.com/in/stylios-lefkopoulos-81b007a0](https://www.linkedin.com/in/stylios-lefkopoulos-81b007a0)

Reviewers' Comments:

Reviewer #1:

Remarks to the Author:

In this manuscript, the authors have provided a single-cell transcriptomic atlas of three amnion samples between 6-8 weeks in the first trimester of pregnancy, identified amnion epithelial stem cells (AESC), amnion mesenchymal stem cells (AMSCs), and a population of bipotent amnion progenitors that generate both AESCs and AMSCs, and then revealed epithelial, mesenchymal and ectodermal primary

trajectories during amnion development. The main weakness of this study lacks immunostaining evidences to confirm single-cell RNA-Seq results, and only one sample every stage also weak the conclusion of the manuscript. The lack of clear sample collection procedures and the exclusion of contamination from other fetal tissues outside the amnion also pose risks to the results. Besides, cell clusters need to be annotated more accurately. In general, it is an important resource data that can help the field to understand human amnion development during early pregnancy, while I do have concerns for this preliminary work.

1. The amnion is connected to other tissues in early embryonic development. The authors should provide evidences to exclude the contamination of other tissues. In addition, different regions of amnion may exhibit different cell composition. The amnion samples are intact? The authors should clarify them.
2. If immunofluorescence staining could be provided for some important cell population, it is critical to help readers understand the cell subsets annotated by the scRNAseq data in Figure 1d.
3. As the authors mentioned that each timepoint is represented by a single sample, the result in Fig. 1e is inconclusive and misleading. I recommend to remove this result.
4. In lines 153-156, the authors provided some results to show that bipotent progenitors were found to express distinct transcription factor characteristics of both AESCs and AMSCs. Only few genes were shown. How many genes are unique or overlapped by AESC and AMSC?
5. Bipotent progenitors could be further clustered into two subgroups represented by cluster 2 and cluster 3. Fig. 2e, f and Fig. S2h showed that AESC and AMSC come from the different two cluster, respectively, which imply that there may be no any bipotent progenitors that has the ability to give rise to both AESC and AMSC. More evidences are required to demonstrate the bipotent potential.
6. For the presence of ectodermal cells (neural stem cells or progenitors) in amnion, additional evidences, such as immunostaining of specific neural ectodermal markers, are critical because the potential contamination from neural ectodermal tissue in the fetus needs to be excluded.
7. The authors observed the presence of blood and immune cells in the amnion. This result is surprising. Although the authors exclude maternal contamination, as well-known, yolk sac and placental villous tissue can produce blood and immune cells in early fetal stage. How to exclude the possible contamination of blood and immune cells produced by yolk sac and placental villous tissue?
8. The authors observed the presence of hematopoietic stem cells. This is an important finding. Traditionally, it is believed that early embryo lacks of the ability to produce hematopoietic stem cells although blood and immune cells could be produced. The author should clearly clarify the population using more evidences.
9. In the Figure 2j, the staining resolution is too low. GATA3 staining is weird. GATA3 should be located in the nucleus, but I observed that in many cells GATA3 is located in other regions of the cells. Is this the background?

Reviewer #2:

Remarks to the Author:

The manuscript "Atlas of amnion development during the first trimester of human pregnancy" presents a comprehensive and detailed single-cell transcriptomic analysis of the human amnion, identifying thirteen distinct cell types and elucidating their developmental trajectories, signaling patterns, and potential roles in immunomodulation during the first trimester of pregnancy. The use of single-cell RNA sequencing to map the cellular landscape of the human amnion during early pregnancy is cutting-edge, providing unprecedented insights into the amnion's cellular composition and dynamics. The identification of cell types, including bipotent amnion progenitors and the delineation of epithelial, mesenchymal, and ectodermal lineages, contributes to the field of developmental biology. The manuscript successfully links cellular composition with functional aspects, such as signaling pathways (e.g., BMP4/BMPR1A, PDGFA/PDGFR α) and potential immunomodulatory roles, offering a foundation for understanding the amnion's roles beyond its physical support to the embryo. The work is significant for its contributions to understanding the complexity and function of the amnion, an essential component of the intrauterine environment, and for potential implications in regenerative medicine and therapeutic applications. There is also limited new mechanistic insight being generated from this newly available dataset. Below are suggestions and comments that need to be addressed :

Major Revisions:

1. While the identification of signaling pathways and cell types provides a framework for understanding amnion development, experimental validation of these functional roles is critical. Specifically, *in vitro* or *in vivo* experiments demonstrating the roles of identified signaling pathways (e.g., BMP4/BMPR1A, PDGFA/PDGFR α) in amnion development and function would greatly strengthen the manuscript's conclusions.
2. The manuscript should more explicitly compare and contrast its findings with existing literature on amnion development and cellular composition (PMID: PMID: 35439430). This includes detailing how the identified cell types and signaling pathways align with or expand upon previous studies. Such a comparison would highlight the novel contributions of this research more clearly. Also including DATASET at earlier stages such as CS7 human datasets and also CS8-11 monkey ones could help to better illustrating amnion development.
3. Expand the discussion on the potential implications of this research for regenerative medicine and therapeutic applications. This should include hypothetical applications based on the identified cell types and signaling pathways, as well as suggestions for future research that could bridge the gap between these findings and clinical applications.

Minor Revisions:

1. Ensure that all methodological details are clearly presented to allow for reproducibility. This includes more explicit descriptions of single-cell RNA sequencing protocols, data processing, and analysis pipelines. Additionally, confirm that all raw and processed data are accessible through public repositories.
2. Ensure that all figures and tables are clearly labeled and referenced in the text. Check the manuscript for any typographical or grammatical errors to maintain a high level of clarity and professionalism.
3. Update the references section to ensure that all cited work is accurately represented and that any recent relevant studies (up to the

manuscript submission date) are included. This will help contextualize the study within the current state of the field.

Reviewer #3:

Remarks to the Author:

Dear Authors,

The manuscripts consist of the single cell transcriptomics analysis of 3 human amnion samples of week 6-8 post conception (6-8WPC). The cell types present in the human amnion of this development age have not been previously analyzed at the single cell level. However, the data provided (donors and age) is rather limited in scope and the validation is poor. Moreover, to my knowledge, the cell types observed are not expected on the amnion at 6-8WPC (although they will be in the chorionamniotic membrane at birth). Hence, I am rather puzzled by the results (could there be yolk sac contamination for example?) and I think some caution paired with more robust validation is necessary to support the (surprising) data. Together, the interpretation of the results and robustness of the claims need to be more strongly supported and the work has a preliminary character. Collaboration with UK tissue bank or similar that could provide this type of samples for analysis and should be considered to increase the robustness.

Major limitations

-The ages analyzed are very close together and it would be more appropriate to consider a wider spread of embryonic ages to generate a bonfide atlas, this seems more a single age group (6-8WPC), there is no separated analysis by sex and the n used is a bare minimum (n=3). There is no positional annotation regarding the place of the amnion samples used (was this the entire amnion or the part covering the anterior embryo or the posterior embryo?). The manuscript would have benefited from including the umbilical cord as that extraembryonic mesoderm may be comparable to the mesoderm covering the amnion.

-I think it is possible that the so-called 'amnion progenitor' cluster could be a different population, located at specific place, for example, at the border with the embryo or close to the umbilical cord, that could simply have a mixed signature meso/ecto? The pseudotime analysis is interesting but with only 3 tissue donors of similar age range it may be inconclusive/premature to infer directionality (the change in directionality is interesting but should be more robustly determined). I suggest the inclusion of more individuals of the same age to exclude interdonor variation and making sure the amnion tissue used is from a similar area, as regional differences may exist and be confounding the data. Hence to annotate as 'progenitor' is premature.

-Direct comparison with the Tyser data that included the amnion (annotated as non-neural ectoderm & advanced mesoderm) should be provided to aid in the improvement of the cell type annotation of the Tyser data; and to generate a developmental trajectory. I would also request the integration (or at least a comparison) of the human data with comparable stages of amnion development from the monkey, as this data are well annotated and should be available for direct comparison and/or integration.

-It would be very important to perform robust validation using IF. The work includes two IF and more (all) cell types should be identified. One of the two AB used GATA3 is not present in the bioinformatics analysis, hence it is unclear why other markers were not used to validate the expected cell types in the amnion. The 6-8WPC embryo is a large membrane and can be used for many stainings. I am sure the labs involved have the ABs for many of the markers identified, such as SOX2, TUBB3 or panKRT.

-The amnion at 6-8WPC is a transparent layer containing no erythroblasts or erythroid or any type of vasculature and to my knowledge no smooth muscle cells or fibroblast or endothelial cells (there is no vasculature).

This is very different from the situation at term, as the avascular amnion has fused with the vascularised chorion/placental plate. The cell types identified seem to come from term amnion and seem unrelated to the fetal state? It is quite puzzling that so many different cell types were observed in such young amnions. These cell types are not expected to be present in the 6-8WPC amnion as such. Hence, providing robust validation is crucial.

I am not sure why this is the case but I would advise extreme caution, as there may be issues with the purity of the samples used or perhaps yolk sac was included as well, as that tissue is vascularized and would contain fetal blood. It would therefore be very important to include images and or histology of the actual (or comparable) tissue used before it went for sequencing. Again, including more extensive IF is crucial to reassure and validate the cell types described, which are unexpected.

-It is good that the authors separated the cells by genotype to exclude that some of the cell types encountered are indeed from maternal origin, such as immune cells. However, I would have preferred to see this presented at the start of the work, when QC is discussed, and information about number and cell type of maternal cells (mostly maternal blood cells I assume) should be explicitly mentioned and stratification between the genotypes of embryo and mother should be provided. During the abortion procedure there is obvious extreme contamination with maternal blood and I am not sure how confident the authors are that the material used for sequence was sufficiently washed from (maternal and fetal) blood and no yolk sac was present. For example, Fig5i seem to be contaminated with (maternal and fetal) blood. How has this been assessed and or prevented? The amnion is a smooth membrane without any blood cells attached, so Fig5i suggests contamination with fetal or maternal blood.

In case the presence is genuine in the samples analysed, do the authors expect cells from the maternal blood to cross the placenta and contact the 6-8WPC amnion, meaning immune cells would be in the exocoelomic space or in the amniotic fluid at 6-8WPC? This would be unexpected/interesting because there is no maternal blood in the intervillous space yet at 6-7WPC.

-Were doublets removed? It is unclear from the quality control, but could justify cells with a double signature, here labeled as bipotent amnion progenitors.

-Fig 1a: at 2 weeks both the amniotic ectoderm (green) as the yolk sac endoderm (blue) is surrounded by a layer of extraembryonic mesoderm (red); comparable with the chorion/TE-derived cells (orange) that are in contact with a layer of extraembryonic mesoderm (red). The umbilical cord structure should also be present. At 8WPC, the amnion is not yet fused with the chorion/placenta and should have its own red cells, separated from the chorion.

Methods should be written concisely, but should contain all elements necessary to allow interpretation and replication of the results. As a guideline, Methods sections typically do not exceed 3,000 words. The Methods should be divided into subsections listing reagents and techniques. When citing previous methods, accurate references should be provided and any alterations should be noted. Information must be provided about: antibody dilutions, company names, catalogue numbers and clone numbers for monoclonal antibodies; sequences of RNAi and cDNA probes/primers or company names and catalogue numbers if reagents are commercial; cell line names, sources and information on cell line identity and authentication. Animal studies and experiments involving human subjects must be reported in detail, identifying the committees approving the protocols. For studies involving human subjects/samples, a statement must be included confirming that informed consent was obtained. Statistical analyses and information on the reproducibility of experimental results should be provided in a section titled "Statistics and Reproducibility".

All Nature Cell Biology manuscripts submitted on or after March 21 2016 must include a Data availability statement at the end of the Methods section. For Springer Nature policies on data availability see <http://www.nature.com/authors/policies/availability.html>; for more information on this particular policy see <http://www.nature.com/authors/policies/data/data-availability-statements-data-citations.pdf>. The Data availability statement should include:

- Accession codes for primary datasets (generated during the study under consideration and designated as "primary accessions") and secondary datasets (published datasets reanalysed during the study under consideration, designated as "referenced accessions"). For primary accessions data should be made public to coincide with publication of the manuscript. A list of data types for which submission to community-endorsed public repositories is mandated (including sequence, structure, microarray, deep sequencing data) can be found here <http://www.nature.com/authors/policies/availability.html#data>.
- Unique identifiers (accession codes, DOIs or other unique persistent identifier) and hyperlinks for datasets deposited in an approved

repository, but for which data deposition is not mandated (see here for details <http://www.nature.com/sdata/data-policies/repositories>).

- At a minimum, please include a statement confirming that all relevant data are available from the authors, and/or are included with the manuscript (e.g. as source data or supplementary information), listing which data are included (e.g. by figure panels and data types) and mentioning any restrictions on availability.
- If a dataset has a Digital Object Identifier (DOI) as its unique identifier, we strongly encourage including this in the Reference list and citing the dataset in the Methods.

We recommend that you upload the step-by-step protocols used in this manuscript to the Protocol Exchange. More details can found at www.nature.com/protocolexchange/about.

All imaging data should be accompanied by scale bars, which should be defined in the legend.

Cropped images of gels/blots are acceptable, but need to be accompanied by size markers, and to retain visible background signal within the linear range (i.e. should not be saturated). The boundaries of panels with low background have to be demarked with black lines. Splicing of panels should only be considered if unavoidable, and must be clearly marked on the figure, and noted in the legend with a statement on whether the samples were obtained and processed simultaneously. Quantitative comparisons between samples on different gels/blots are discouraged; if this is unavoidable, it should only be performed for samples derived from the same experiment with gels/blots were processed in parallel, which needs to be stated in the legend.

- For line art, graphs, charts and schematics we prefer Adobe Illustrator (.AI), Encapsulated PostScript (.EPS) or Portable Document Format (.PDF). Files should be saved or exported as such directly from the application in which they were made, to allow us to restyle them according to our journal house style.
- We accept PowerPoint (.PPT) files if they are fully editable. However, please refrain from adding PowerPoint graphical effects to objects, as this results in them outputting poor quality raster art. Text used for PowerPoint figures should be Helvetica (preferred) or Arial.
- We do not recommend using Adobe Photoshop for designing figures, but we can accept Photoshop generated (.PSD or .TIFF) files only if each element included in the figure (text, labels, pictures, graphs, arrows and scale bars) are on separate layers. All text should be editable in 'type layers' and line-art such as graphs and other simple schematics should be preserved and embedded within 'vector smart objects' - not flattened raster/bitmap graphics.
- Some programs can generate Postscript by 'printing to file' (found in the Print dialogue). If using an application not listed above, save the file in PostScript format or email our Art Editor, Allen Beattie for advice (a.beattie@nature.com).

SUPPLEMENTARY INFORMATION – Supplementary information is material directly relevant to the conclusion of a paper, but which cannot be included in the printed version in order to keep the manuscript concise and accessible to the general reader. Supplementary information is an integral part of a Nature Cell Biology publication and should be prepared and presented with as much care as the main display item, but it must not include non-essential data or text, which may be removed at the editor's discretion. All supplementary

material is fully peer-reviewed and published online as part of the HTML version of the manuscript. Supplementary Figures and Supplementary Notes are appended at the end of the main PDF of the published manuscript.

The total number of Supplementary Figures (not including the “unprocessed scans” Supplementary Figure) should not exceed the number of main display items (figures and/or tables (see our Guide to Authors and March 2012 editorial <http://www.nature.com/ncb/authors/submit/index.html#suppinfo>; <http://www.nature.com/ncb/journal/v14/n3/index.html#ed>). No restrictions apply to Supplementary Tables or Videos, but we advise authors to be selective in including supplemental data.

GUIDELINES FOR EXPERIMENTAL AND STATISTICAL REPORTING

REPORTING REQUIREMENTS – To improve the quality of methods and statistics reporting in our papers we have recently revised the reporting checklist we introduced in 2013. We are now asking all life sciences authors to complete two items: an Editorial Policy Checklist (found here <https://www.nature.com/authors/policies/Policy.pdf>) that verifies compliance with all required editorial policies and a reporting summary (found here <https://www.nature.com/authors/policies/ReportingSummary.pdf>) that collects information on experimental design and reagents. These documents are available to referees to aid the evaluation of the manuscript. Please note that these forms are dynamic ‘smart pdfs’ and must therefore be downloaded and completed in Adobe Reader. We will then flatten them for ease of use by the reviewers. If you would like to reference the guidance text as you complete the template, please access these flattened versions at <http://www.nature.com/authors/policies/availability.html>.

We strongly recommend the presentation of source data for graphical and statistical analyses as a separate Supplementary Table, and request that source data for all independent repeats are provided when representative experiments of multiple independent repeats, or averages of two independent experiments are presented. This supplementary table should be in Excel format, with data for different figures provided as different sheets within a single Excel file. It should be labelled and numbered as one of the supplementary tables, titled “Statistics Source Data”, and mentioned in all relevant figure legends.

Version 1:

Decision Letter:

*Please delete the link to your author homepage if you wish to forward this email to co-authors.

Dear Magda,

As already communicated, one of the reviewers (the original reviewer #2) of your manuscript "Atlas of Amnion Development During the First Trimester of Human Pregnancy" did not provide their report. In the meantime, reviewer #1 provided their report and continued to raise some issues. We therefore had to further discuss the different issues within the referee panel and asked reviewer #3 to provide their view on your responses, not only to their own original comments, but also the original comments by reviewers #1 and #2. Your revised manuscript has therefore been seen by the original reviewers #1 and #3 and their reports, which I have already shared with you, are pasted below. As you saw from their comments, they find the work improved, but have raised some important concerns, which is why we requested your response to them before proceeding with making a decision.

Taking everything into consideration, including your response to the reviewer comments, we are willing to allow one more round of revision to address the referees' concerns. In particular, it would be essential to address all the points raised by the reviewers, including with new data, as you have proposed in your point-by-point response that you shared with us via e-mail. If the referees continue to be unconvinced after revision, we would unfortunately not be able to pursue the study further for this journal. Furthermore, when you submit your revised manuscript, we request that you please provide a marked up version of the manuscript file, so that it is easier for the reviewers to assess the additions/changes compared to the previous version they have seen.

Finally, please pay close attention to our guidelines on statistical and methodological reporting (listed below) as failure to do so may delay the reconsideration of the revised manuscript. In particular please provide:

- a Supplementary Figure including unprocessed images of all gels/blots in the form of a multi-page pdf file. Please ensure that blots/gels are labeled and the sections presented in the figures are clearly indicated.
- a Supplementary Table including all numerical source data in Excel format, with data for different figures provided as different sheets within a single Excel file. The file should include source data giving rise to graphical representations and statistical descriptions in the paper and for all instances where the figures present representative experiments of multiple independent repeats, the source data of all repeats should be provided.

We would be happy to consider a revised manuscript that would satisfactorily address these points, unless a similar paper is published elsewhere, or is accepted for publication in Nature Cell Biology in the meantime.

- ensure that it conforms to our format instructions and publication policies (see below and www.nature.com/nature/authors/).
- provide a point-by-point rebuttal to the full referee reports verbatim, as provided at the end of this letter.
- provide the completed Editorial Policy Checklist (found here <https://www.nature.com/authors/policies/Policy.pdf>), and Reporting Summary (found here <https://www.nature.com/authors/policies/ReportingSummary.pdf>). This is essential for reconsideration of the manuscript and these documents will be available to editors and referees in the event of peer review. For more information see <http://www.nature.com/authors/policies/availability.html> or contact me.

Nature Cell Biology is committed to improving transparency in authorship. As part of our efforts in this direction, we are now requesting that all authors identified as 'corresponding author' on published papers create and link their Open Researcher and Contributor Identifier (ORCID) with their account on the Manuscript Tracking System (MTS), prior to acceptance. ORCID helps the scientific community achieve unambiguous attribution of all scholarly contributions. You can create and link your ORCID from the home page of the MTS by clicking on 'Modify my Springer Nature account'. For more information please visit <http://www.springernature.com/orcid>.

Link Redacted

We would like to receive a revised submission within one month, but if you believe you can obtain more data/material (additional to what you have proposed to provide/have already done per the reviewer responses you shared with me via e-mail) and/or you require for any reason additional time, please contact me and I can happily arrange that.

We hope that you will find our referees' comments and editorial guidance helpful. Please do not hesitate to contact me if there is anything you would like to discuss.

Best wishes,

Stelios

Stylios Lefkopoulos, PhD
He/him/his
Senior Editor, Nature Cell Biology
Springer Nature
Heidelberger Platz 3, 14197 Berlin, Germany

E-mail: stylios.lefkopoulos@springernature.com
Twitter: @s_lefkopoulos
LinkedIn: [linkedin.com/in/stylios-lefkopoulos-81b007a0](https://www.linkedin.com/in/stylios-lefkopoulos-81b007a0)

Reviewers' Comments:

Reviewer #1 (Remarks to the Author):

The reversion has addressed my main concerns. However, I have some concerns:

1. In the Epi-trajectory, the author defined "the other from Epi-progenitor1 to amnion epithelial cells and then to macrophages". The developmental origin of macrophage from the amnion is very unexpected. However, the conclusion was obtained solely based on the developmental trajectory. The result is very primarily. It is required to address the issue by functional assays using stem cells.
2. The author defined a population as amnion ectodermal cells or neural-like cells by expression of TUBB3 and Sox2 in Figure 2C. Any alone marker is not specific for ectodermal cells or neural-like cells. Do the authors check the double-positive cells of Sox2 and TUBB3? TUBB3 is sometimes expressed by fibroblast cells. It is required to provide double-staining of more markers or deeply analyze the gene expression profiles for this population.
3. In the Figure 4A, it is very weird that the data from different sources are completely separated without any overlap. Is this caused by batch effects? The author claimed "the various in vitro amnion models represent different developmental stages in vivo". Is there a continuous correlation among them in developmental trajectory?

Reviewer #3 (Remarks to the Author):

Dear authors,

The manuscript has increased considerably in quality and robustness.

You increased the N, the quality control of the data improved significantly, the integration with other datasets was performed, the validation is not very robust but there was a effort to address the suggestion.

I was further asked to cross-comment on the authors' rebuttal to original reviewers #1 and #2.

Regarding reviewer #1:

1. The authors provide images and are doing an effort, but not really addressing the issue. Perhaps the authors should mention the aspect of the different regions the Discussion?
2. The authors did some effort to include some validation. Not great, but sufficient, although I cannot recognise the amniotic structure and bilayer structure of the membrane. This is actually important to point 1.6 (see 1.6).
3. Ok
4. The additional analysis is adequate providing extra markers between the populations
5. The names used 'epi-progenitors' and 'Mes-progenitors' are quite ambiguous and could be replaced by the used terminology 'AESC' and 'AMSC'? This is used by reviewer #2 to avoid unambiguity between 'ectoderm' and 'epithelium', etc. One could consider
'Epi-progenitor_1' = 'AESC_progenitor_1'
'Amnion-Epi' = 'AESC'
'Mes-progenitors' = 'AMSC_progenitor'
'Amnion-Mes' = 'AMSC'
6. I dont recognise the amniotic structure as a bilayer structure. I am also not able to see whether the SOX2+ cells could indeed be of contamination from neural ectodermal tissue. Higher quality images could be provided, lower magnifications and proximity to the embryo. I think this would need to be clarified further to exclude contamination or at least mention in Discussion. Also understanding whether the ectoderm cells would be close to AESC or AMSC or in a specific location (connects to point 1.1) would be valuable to know/discuss.
7. The authors removed the part of the blood.
8. The authors removed the part of the blood.
9. The IF is of poor quality. This could be improved to add to the robustness of the manuscript.

Regarding reviewer #2:

- 2.1 The authors followed the reviewer's suggestion.
- 2.2 The authors followed the reviewer's suggestion.
- 2.3 The authors followed the reviewer's suggestion.
- 2.4 The authors made an effort to clarify methodology and include GEO number.
- 2.5 'Ensure that all figures and tables are clearly labeled and referenced in the text. Check the manuscript for any typographical or grammatical errors to maintain a high level of clarity and professionalism.' It is very difficult to understand what/where was modified without having a marked-up manuscript.
I assume the authors went through the English throughout and improved the English as requested.
- 2.6 Ensure correct references are used: See 2.5. It is difficult for us to directly compare the references used between the new and original manuscript. It would be much more convenient to have a manuscript with clear modifications to easily understand what was changed exactly.
However, the authors made an effort to include more recent references as requested.

TITLE – should be no more than 100 characters including spaces, without punctuation and avoiding technical terms, abbreviations, and active verbs.

Methods should be written concisely, but should contain all elements necessary to allow interpretation and replication of the results. As a guideline, Methods sections typically do not exceed 3,000 words. The Methods should be divided into subsections listing reagents and techniques. When citing previous methods, accurate references should be provided and any alterations should be noted. Information must be provided about: antibody dilutions, company names, catalogue numbers and clone numbers for monoclonal antibodies; sequences of RNAi and cDNA probes/primers or company names and catalogue numbers if reagents are commercial; cell line names, sources and information on cell line identity and authentication. Animal studies and experiments involving human subjects must be reported in detail, identifying the committees approving the protocols. For studies involving human subjects/samples, a statement must be included confirming that informed consent was obtained. Statistical analyses and information on the reproducibility of experimental results should be provided in a section titled "Statistics and Reproducibility".

All Nature Cell Biology manuscripts submitted on or after March 21 2016 must include a Data availability statement at the end of the Methods section. For Springer Nature policies on data availability see <http://www.nature.com/authors/policies/availability.html>; for more information on this particular policy see <http://www.nature.com/authors/policies/data/data-availability-statements-data-citations.pdf>. The Data availability statement should include:

- Accession codes for primary datasets (generated during the study under consideration and designated as "primary accessions") and secondary datasets (published datasets reanalysed during the study under consideration, designated as "referenced accessions"). For primary accessions data should be made public to coincide with publication of the manuscript. A list of data types for which submission to community-endorsed public repositories is mandated (including sequence, structure, microarray, deep sequencing data) can be found here <http://www.nature.com/authors/policies/availability.html#data>.
- Unique identifiers (accession codes, DOIs or other unique persistent identifier) and hyperlinks for datasets deposited in an approved repository, but for which data deposition is not mandated (see here for details <http://www.nature.com/sdata/data-policies/repositories>).
- At a minimum, please include a statement confirming that all relevant data are available from the authors, and/or are included with the manuscript (e.g. as source data or supplementary information), listing which data are included (e.g. by figure panels and data types) and mentioning any restrictions on availability.
- If a dataset has a Digital Object Identifier (DOI) as its unique identifier, we strongly encourage including this in the Reference list and

citing the dataset in the Methods.

We recommend that you upload the step-by-step protocols used in this manuscript to protocols.io. More details can found at <https://www.protocols.io/help/publish-articles>.

All imaging data should be accompanied by scale bars, which should be defined in the legend.

Cropped images of gels/blots are acceptable, but need to be accompanied by size markers, and to retain visible background signal within the linear range (i.e. should not be saturated). The boundaries of panels with low background have to be demarked with black lines. Splicing of panels should only be considered if unavoidable, and must be clearly marked on the figure, and noted in the legend with a statement on whether the samples were obtained and processed simultaneously. Quantitative comparisons between samples on different gels/blots are discouraged; if this is unavoidable, it should only be performed for samples derived from the same experiment with gels/blots were processed in parallel, which needs to be stated in the legend.

The total number of Supplementary Figures (not including the “unprocessed scans” Supplementary Figure) should not exceed the number of main display items (figures and/or tables (see our Guide to Authors and March 2012 editorial <http://www.nature.com/ncb/authors/submit/index.html#suppinfo>; <http://www.nature.com/ncb/journal/v14/n3/index.html#ed>). No restrictions apply to Supplementary Tables or Videos, but we advise authors to be selective in including supplemental data.

GUIDELINES FOR EXPERIMENTAL AND STATISTICAL REPORTING

REPORTING REQUIREMENTS – To improve the quality of methods and statistics reporting in our papers we have recently revised the reporting checklist we introduced in 2013. We are now asking all life sciences authors to complete two items: an Editorial Policy Checklist (found here <https://www.nature.com/authors/policies/Policy.pdf>) that verifies compliance with all required editorial policies and a reporting summary (found here <https://www.nature.com/authors/policies/ReportingSummary.pdf>) that collects information on experimental design and reagents. These documents are available to referees to aid the evaluation of the manuscript. Please note that these forms are dynamic ‘smart pdfs’ and must therefore be downloaded and completed in Adobe Reader. We will then flatten them for ease of use by the reviewers. If you would like to reference the guidance text as you complete the template, please access these flattened versions at <http://www.nature.com/authors/policies/availability.html>.

STATISTICS – Wherever statistics have been derived the legend needs to provide the n number (i.e. the sample size used to derive statistics) as a precise value (not a range), and define what this value represents. Error bars need to be defined in the legends (e.g. SD, SEM) together with a measure of centre (e.g. mean, median). Box plots need to be defined in terms of minima, maxima, centre, and percentiles. Ranges are more appropriate than standard errors for small data sets. Wherever statistical significance has been derived, precise p values need to be provided and the statistical test used needs to be stated in the legend. Statistics such as error bars must not be derived from n<3. For sample sizes of n<5 please plot the individual data points rather than providing bar graphs. Deriving statistics from technical replicate samples, rather than biological replicates is strongly discouraged. Wherever statistical significance has been derived, precise p values need to be provided and the statistical test stated in the legend.

We strongly recommend the presentation of source data for graphical and statistical analyses as a separate Supplementary Table, and request that source data for all independent repeats are provided when representative experiments of multiple independent repeats, or averages of two independent experiments are presented. This supplementary table should be in Excel format, with data for different figures provided as different sheets within a single Excel file. It should be labelled and numbered as one of the supplementary tables, titled “Statistics Source Data”, and mentioned in all relevant figure legends.

Version 2:

Decision Letter:

16th March 2025

Dear Magda,

Thank you for submitting your revised manuscript "Atlas of Amnion Development During the First Trimester of Human Pregnancy" (NCB-RS53598B). As you know it was sent back to the original referees #1 and #3 and, while referee #3 signed off, referee #1 continued to raise a couple of points, which we shared with you. Following your response to these comments via e-mail, which you gave us permission to share with the reviewers, we then discussed these issues with both referee #1 and #3. Ultimately, the reviewers find that the paper has overall improved in revision and the remaining issues should and can be textually addressed. Therefore, we'll be happy in principle to publish your study in Nature Cell Biology, pending minor revisions to satisfy the referees' final requests and to comply with our editorial and formatting guidelines.

If the current version of your manuscript is in a PDF format, please email us a copy of the file in an editable format (Microsoft Word or LaTeX)-- we cannot proceed with PDFs at this stage.

Thank you again for your interest in Nature Cell Biology. Please do not hesitate to contact me if you have any questions.

Best wishes,
Stelios

Stylianos Lefkopoulos, PhD
He/him/his
Senior Editor, Nature Cell Biology
Springer Nature
Heidelberger Platz 3, 14197 Berlin, Germany

E-mail: stylianos.lefkopoulos@springernature.com
Twitter: @s_lefkopoulos
LinkedIn: [linkedin.com/in/stylianos-lefkopoulos-81b007a0](https://www.linkedin.com/in/stylianos-lefkopoulos-81b007a0)

Reviewer #1 (Remarks to the Author):

Although the revised manuscript addressed my main concerns, some images are still unclear. For example, in Extended Figure 4e. Double stainings of SOX2 and TUBB3 were very unconvincing. SOX2 is a nuclear protein and TUBB3 is a cytoskeletal protein. However, SOX2 and TUBB3 double-positive cells look like false-positive cells. Deep analysis of the gene expression profiles for this population is still required to demonstrate the presence of this population. In Extended Figure, CD45 staining is weird. CD45 is a cell membrane protein. A higher magnification image is required to assure the staining.

ADDITIONAL COMMENTS TO THE AUTHORS' RESPONSE TO THE COMMENTS ABOVE

Although their analysis in the cynomolgus monkey scRNA-seq data are not fully supporting their conclusion, we agree and suggest that the authors weaken their conclusion in the article. I have no other question.

Reviewer #3 (Remarks to the Author):

Dear authors,

The manuscript has significantly increased in quality, robustness and clarity during the review process.

ADDITIONAL COMMENTS ON THE AUTHORS' RESPONSE TO THE REMAINING POINTS BY REVIEWER #1

The amnion is known for its robust/strong structure, not for its fragility. Hence I am surprised the authors are not able to provide higher magnifications of immunostainings performed recently. This just adds to the concerns regarding validation on the population of interest. Adding sc-RNA from monkey will be informative, but does not replace the validation of the cells in human. Hence it would be advised indeed to downplay the claims and suggesting further investigation.

Version 3:

Decision Letter:

Dear Magda,

I am pleased to inform you that your manuscript, "Atlas of Amnion Development During the First Trimester of Human Pregnancy", has now been accepted for publication in Nature Cell Biology - congratulations!

If you have any questions about our publishing options, costs, Open Access requirements, or our legal forms, please contact

ASJournals@springernature.com

Please note that *Nature Cell Biology* is a Transformative Journal (TJ). Authors may publish their research with us through the traditional subscription access route or make their paper immediately open access through payment of an article-processing charge (APC). Authors will not be required to make a final decision about access to their article until it has been accepted. [Find out more about Transformative Journals](https://www.springernature.com/gp/open-research/transformative-journals)

Authors may need to take specific actions to achieve [compliance with funder and institutional open access mandates](https://www.springernature.com/gp/open-research/funding/policy-compliance-faqs). If your research is supported by a funder that requires immediate open access (e.g. according to [Plan S principles](https://www.springernature.com/gp/open-research/plan-s-compliance)) then you should select the gold OA route, and we will direct you to the compliant route where possible. For authors selecting the subscription publication route, the journal's standard licensing terms will need to be accepted, including [self-archiving policies](https://www.springernature.com/gp/open-research/policies/journal-policies). Those licensing terms will supersede any other terms that the author or any third party may assert apply to any version of the manuscript.

If you have not already done so, we strongly recommend that you upload the step-by-step protocols used in this manuscript to protocols.io (<https://protocols.io>), an open online resource that allows researchers to share their detailed experimental know-how. All uploaded protocols are made freely available and are assigned DOIs for ease of citation. Protocols and Nature Portfolio journal papers in which they are used can be linked to one another, and this link is clearly and prominently visible in the online versions of both. Authors who performed the specific experiments can act as primary authors for the Protocol as they will be best placed to share the methodology details, but the Corresponding Author of the present research paper should be included as one of the authors. By uploading your Protocols onto protocols.io, you are enabling researchers to more readily reproduce or adapt the methodology you use, as well as increasing the visibility of your protocols and papers. You can also establish a dedicated workspace to collect your lab Protocols. Further information can be found at <https://www.protocols.io/help/publish-articles>.

Nature Cell Biology encourages authors presenting evidence for cell, biological, molecular, and genetic interactions to consider communicating these findings using Biofactoid (<https://biofactoid.org/>). This tool helps users share a searchable representation of interactions (e.g. binding, gene expression, post-translational modification) between genes, gene products, or chemicals. Information added to Biofactoid, with author attribution, is shared on social media and public databases, such as Pathway Commons, where it can be discovered and analyzed in the context of a large and growing corpus of knowledge.

With kind regards,
Stelios

Stylianos Lefkopoulos, PhD
He/him/his
Senior Editor, Nature Cell Biology
Springer Nature
Heidelberger Platz 3, 14197 Berlin, Germany

E-mail: stylianos.lefkopoulos@springernature.com
Bluesky: [@slefkopoulos.bsky.social](https://bsky.app/profile/@slefkopoulos.bsky.social)
LinkedIn: www.linkedin.com/in/stylianos-lefkopoulos-81b007a0

Click here if you would like to recommend Nature Cell Biology to your librarian

<http://www.nature.com/subscriptions/recommend.html#forms>

** Visit the Springer Nature Editorial and Publishing website at http://editorial-jobs.springernature.com?utm_source=ejP_NCB_email&utm_medium=ejP_NCB_email&utm_campaign=ejp_NCB for more information about our career opportunities. If you have any questions please click [here](mailto:editorial.publishing.jobs@springernature.com).
